# Automated Dynamic Mechanism Design

**Hanrui Zhang**
Duke University
hrzhang@cs.duke.edu

**Vincent Conitzer**
Duke University
conitzer@cs.duke.edu

## Abstract

We study Bayesian automated mechanism design in unstructured dynamic environments, where a principal repeatedly interacts with an agent, and takes actions based on the strategic agent's report of the current state of the world. Both the principal and the agent can have arbitrary and potentially different valuations for the actions taken, possibly also depending on the actual state of the world. Moreover, at any time, the state of the world may evolve arbitrarily depending on the action taken by the principal. The goal is to compute an optimal mechanism which maximizes the principal's utility in the face of the self-interested strategic agent.

We give an efficient algorithm for computing optimal mechanisms, with or without payments, under different individual-rationality constraints, when the time horizon is constant. Our algorithm is based on a sophisticated linear program formulation, which can be customized in various ways to accommodate richer constraints. For environments with large time horizons, we show that the principal's optimal utility is hard to approximate within a certain constant factor, complementing our algorithmic result. These results paint a relatively complete picture for automated dynamic mechanism design in unstructured environments. We further consider a special case of the problem where the agent is myopic, and give a refined efficient algorithm whose time complexity scales linearly in the time horizon.

In the full version of the paper, we show that memoryless mechanisms, which are without loss of generality optimal in Markov decision processes without strategic behavior, do not provide a good solution for our problem, in terms of both optimality and computational tractability. Moreover, we present experimental results where our algorithms are applied to synthetic dynamic environments with different characteristics, which not only serve as a proof of concept for our algorithms, but also exhibit intriguing phenomena in dynamic mechanism design.

## 1 Introduction

Consider the following scenario. A company assembles an internal research group to develop key technologies to be used in the company's next-generation product in 5 years. The more progress the group makes, the more successful the product is likely to be. Since research progress is hard to monitor, the company manages the group based on its annual reports. At the beginning of each year, the group submits a report, summarizing its progress in the preceding year, as well as its needs for the current year. Taking into consideration this report (and possibly also reports from previous years), the company then decides the compensation level and the headcount of the group in the current year. Moreover, after the product launches, the company may also pay a bonus to members of the group, depending on how successful the product is.

For simplicity, suppose an annual report consists of two items: research progress (satisfactory/unsatisfactory), and need to expand (no request/request for an intern/request for a full-time employee). The company's goal is to encourage and facilitate research progress while keeping the expenses reasonable. So, a natural managing strategy is to increase (resp. decrease) the compensation

35th Conference on Neural Information Processing Systems (NeurIPS 2021).

level when the reported research progress is satisfactory (resp. unsatisfactory), and to allow the group to expand only when necessary, i.e., when the reported research progress is unsatisfactory. However, the research group may have a different goal than the company's. Suppose members of the group do not care about the success of the product *per se*. Instead, their primary goal is to maximize the total compensation received from the company, and for this reason, they may be incentivized to *misreport* the situation. In other words, the company faces a *dynamic mechanism design* problem, where the *principal* (i.e., the company) needs to implement (and commit to) a *mechanism* (i.e., a managing strategy) that achieves its goal through *repeated* interactions, in the presence of strategic behavior of the *agent* (i.e., the research group).

Indeed this problem is nontrivial. For example, if the company implements the above strategy, then the group will report satisfactory progress regardless of the actual situation, which maximizes the group's total compensation over the 5 years, but also causes greater expenses for the company and jeopardizes the success of the product. To counter this, the company may additionally promise a significant bonus contingent on the success of the product. This creates incentives for the group to make more progress, and discourages overreporting the progress, because the group is not allowed to expand when the reported progress is satisfactory. That is, if actual progress is unsatisfactory, this introduces an incentive to report this truthfully so that the group may expand. However, this also runs the risk of encouraging the group to report unsatisfactory progress in order to expand even if actual progress is satisfactory, because more members always make more progress, which leads to a higher (chance of) bonus, whereas the cost of expanding is paid by the company and therefore irrelevant to the group.

One may try to fix this by introducing more rules, possibly replacing existing ones. For example, the company may allow the group to recruit an intern, but not a full-time employee, when the reported progress is unsatisfactory. Then, in the next year, if the reported progress improves, the company allows the group to make a return offer to the intern as a full-time employee. Or alternatively, the company may unconditionally allow the group to recruit interns (which are less costly), but never full-time employees. In addition to the above, the company could also temporarily decrease the compensation level when a new member joins, and later adjust the compensation based on how the reported progress improves. While all these ad hoc rules make intuitive sense, it is not immediately clear which (combinations of) rules are better, how to optimize parameters of these rules (e.g., the number of new members allowed per year and the amount by which the compensation is adjusted), or whether there is a better set of rules that look totally different.

As demonstrated by the foregoing discussion, in general, the problem of finding an optimal mechanism in *unstructured* dynamic environments, such as the above example, turns out to be extremely rich and challenging. In such environments, the actions of the principal may go beyond the allocation of items to the agent, and affect the state of the world in arbitrary ways. Moreover, both the principal and the agent may have arbitrary valuations for these actions, which also depend on the current state of the world. In economic theory, the *characterize-and-solve* approach [25, 13, 30] to mechanism design has achieved spectacular success in both static and dynamic environments, by exploiting structure of the environment to construct a characterization of optimal mechanisms, often leading to closed-form or computationally tractable solutions. However, since the environments under consideration here are loosely structured at best, the classical characterize-and-solve approach does not seem particularly suited. When disregarding the agent's incentives, one could treat the problem of finding an optimal strategy as a *planning* problem, which is known to be solvable efficiently [6, 21, 31]. However, as discussed above, the agent's strategic behavior can ruin the performance of such a strategy. From a computational perspective, while numerous methods for *automated mechanism design*, which efficiently compute optimal mechanisms without heavily exploiting structures of the environment, have been proposed [10, 11, 12], all existing methods work only for static environments with one-time interactions, and it is not immediately clear how to generalize these methods to dynamic environments. All this brings us to the following question:

> *Can we efficiently compute optimal mechanisms in unstructured dynamic environments?*

## 1.1  Our Results

In this paper, we study the problem of computing optimal mechanisms in *single-agent*, *discrete-time* dynamic environments with a *finite time horizon*, without any further structural assumptions. Our main results (presented in Section 3) can be summarized as follows:

- **Efficient algorithm**: when the time horizon is fixed, there is a polynomial-time algorithm for computing optimal mechanisms, with or without payments, that maximize the principal's utility facing a strategic agent.

- **Inapproximability**: when the time horizon can be large, it is NP-hard to approximate the principal's optimal utility within a factor of $(7/8 + \varepsilon)$ for any $\varepsilon > 0$.

To the best of our knowledge, our algorithm for constant time horizons is the first that efficiently computes optimal mechanisms in unstructured dynamic environments. The fact that our algorithm cannot scale beyond constant time horizons is by no means surprising: optimal dynamic mechanisms generally depend on the entire history, and as a result, the straightforward description of such a mechanism is exponentially large in the time horizon. Our inapproximability result further rules out the possibility of computing succinct representations of approximately optimal mechanisms that can be efficiently evaluated. These results together paint a complete picture of the computational complexity of dynamic mechanism design in unstructured environments.

## 1.2  Further Related Work

**Dynamic mechanism design.**    The problem we study can be situated in the broad area of dynamic mechanism design, and below we discuss some representative related work. For a more comprehensive exposition, see, e.g., the survey by Pavan [29] and the one by Bergemann and Välimäki [8]. In the context of efficient (i.e., welfare-maximizing) mechanisms, Bergemann and Välimäki [7] propose the dynamic pivot mechanism, which generalizes the VCG mechanism in static environments, and Athey and Segal [2] propose the team mechanism, which focuses on budget-balancedness. As for optimal (i.e., revenue-maximizing) mechanisms, which are more closely related to our results, following earlier work [5, 13, 17], Pavan et al. [30] generalize the classical characterization by Myerson [25] into dynamic environments, unifying previous results with continuous type spaces. Ashlagi et al. [1] study ex-post individual-rational dynamic mechanisms for repeated auctions, and give an efficient $(1 - \varepsilon)$-approximation to the optimal revenue for a single agent with independent valuations across items. Mirrokni et al. [24] study non-clairvoyant dynamic mechanism design, where future distributional knowledge is unavailable to the principal. All these results for optimal mechanisms follow the characterize-and-solve approach, which is quite different from the computational approach that we take.

Particularly related to our results is the work by Papadimitriou et al. [28], who study a setting where one item is sold at each time, and agents' valuations can be correlated across items. They show that designing an optimal deterministic mechanism is computationally hard even when there is only one agent and two items (thereby ruling out the possibility of efficiently computing optimal deterministic mechanisms in our model, which is more general). And moreover, they give a polynomial-size linear program formulation for optimal randomized mechanisms for independent agents when the number of agents and the time horizon are both constant. Restricted to a single agent, their LP formulation can be viewed as a special case of our main result: they focus on revenue maximization with a single item to be allocated at each time, in a model where the principal's actions cannot affect the future valuations of the agent; on the other hand, we allow the principal to care about actions as well as revenue, with actions being general and unstructured (as opposed to allocation/no allocation), where the future state of the world can depend arbitrarily on the principal's actions as well as the current state.

**Automated Mechanism Design.**    There is a rich body of research regarding automated mechanism design (AMD) in (essentially) static environments. Conitzer and Sandholm [10, 11] initiated the study of automated mechanism design. They consider various specific static setups, and show that computing optimal deterministic mechanisms, even with a single agent, is often NP-hard (which also rules out the possibility of efficiently computing optimal deterministic mechanisms in our model, since the 1-period case is a special case), while computing optimal randomized mechanisms is often tractable. Conceptually related to our model, Hajiaghayi et al. [20] consider a model where agents

enter and leave the mechanism online (but still have one-time interactions with the mechanism), and provide an algorithm for computing mechanisms that are competitive against the optimal allocation from hindsight. Sandholm et al. [35] study automated design of multistage mechanisms, but these are not for dynamic settings; instead, the motivation is to implement static mechanisms using multiple rounds of queries in order to minimize the communication cost. Sandholm and Likhodedov [34] study automated design of combinatorial auction mechanisms, and Balcan et al. [3, 4] study the sample complexity thereof. Kephart and Conitzer [22, 23] and Zhang et al. [37] study AMD with partial verification and/or reporting costs. More recently, various methods have been proposed for automated mechanism design via machine learning [14, 27], and in particular, deep learning [15, 18, 36, 32]. All these results are essentially for static environments, whereas in this paper, we focus solely on AMD in dynamic environments. Another emerging research direction is Bayesian persuasion in dynamic environments [16, 33]. In particular, Celli et al. [9] study an algorithmic persuasion problem in extensive-form games, where a single signal is sent at the very beginning, and Gan et al. [19] study an algorithmic persuasion problem in infinite-horizon discounted MDPs, where a new signal is sent at every time. These persuasion problems can be viewed as a dual problem of ours: in our problem, the principal has the commitment power, and tries to encourage the agent to truthfully report their private information, whereas in (dynamic) Bayesian persuasion, the agent has the commitment power, and tries to induce the principal to act in favor of the agent by selectively revealing their private information.

## 2 Preliminaries

**Dynamic environments.** Throughout this paper, we consider single-agent, discrete-time environments with a finite time horizon. Below, we give a general definition of such a dynamic environment. Let $T$ be the time horizon, $\mathcal{S}$ be the state space, and $\mathcal{A}$ be the action space. The agent observes the state, but the principal controls the action that is taken. For each $t \in [T]$, let $v_t^P : \mathcal{S} \times \mathcal{A} \to \mathbb{R}$ be the principal's valuation function, where for each state $s \in \mathcal{S}$ and action $a \in \mathcal{A}$, $v_t^P(s, a)$ is the value of the principal when playing action $a$ in state $s$, at time $t$; similarly, let $v_t^A : \mathcal{S} \times \mathcal{A} \to \mathbb{R}$ be the agent's value function. Let $P_0 \in \Delta(\mathcal{S})$ be the initial distribution over states, and for each $s \in \mathcal{S}$, denote by $P_0(s)$ the probability that the initial state is $s$. Moreover, for each $t \in [T]$, let $P_t : \mathcal{S} \times \mathcal{A} \to \Delta(\mathcal{S})$ be the transition operator, which maps a state-action pair $(s, a)$ at time $t$ to the distribution of the next state at time $t + 1$, $P_t(s, a) \in \Delta(\mathcal{S})$. We denote by $P_t(s, a, s')$ the probability that the next state is $s'$ when playing action $a$ in state $s$ at time $t \in [T]$. For notational simplicity, let $P_0(s, a, s') = P_0(s')$ for all $s, s' \in \mathcal{S}$ and $a \in \mathcal{A}$. (Note that the first *actual* action is taken at $t = 1$ — not $t = 0$ — possibly based on the state at $t = 1$.)

**Histories.** A $t$-step history is a sequence of states and actions $(s_1, a_1, s_2, \ldots, a_{t-1}, s_t, a_t)$, where for each $i \in [t]$, it is the case that $s_i \in \mathcal{S}$ and $a_i \in \mathcal{A}$. For each $t \in [T]$, let $\mathcal{H}_t$ be the set of all possible $t$-step histories, i.e.,

$$\mathcal{H}_t = \{(s_1, a_1, \ldots, s_t, a_t) \mid s_i \in \mathcal{S}, a_i \in \mathcal{A} \text{ for all } i \in [t]\}.$$

For each $h = (s_1, a_1, \ldots, s_t, a_t) \in \mathcal{H}$, let $|h| = t$, and moreover, for any $s_{t+1} \in \mathcal{S}$, $a_{t+1} \in \mathcal{A}$, let $h + (s_{t+1}, a_{t+1}) = (s_1, a_1, \ldots, s_{t+1}, a_{t+1})$. Let $\mathcal{H}_0 = \{\emptyset\}$, where $\emptyset$ corresponds to the empty history with $|\emptyset| = 0$. Let $\mathcal{H} = \mathcal{H}_0 \cup \bigcup_{t \in [T-1]} \mathcal{H}_t$ be the set of all possible histories of length at most $T - 1$ in the dynamic environment. Note that $\mathcal{H}$ does not contain histories of length $T$.

**Dynamic mechanisms.** Dynamic mechanisms are more powerful than static ones, in that they may depend on the *entire history*, rather than only the current state. A (randomized) dynamic mechanism $M = (\pi, p)$ consists of an action policy $\pi$ and a payment function $p$. The action policy $\pi : \mathcal{H} \times \mathcal{S} \to \Delta(\mathcal{A})$ maps each history $h \in \mathcal{H}$, extended with the reported current state $s \in \mathcal{S}$, to a distribution over actions $\pi(h, s) \in \Delta(\mathcal{A})$. We denote by $\pi(h, s, a)$ the probability that the action taken by the mechanism is $a$ for $(h, s)$. The payment function $p : \mathcal{H} \times \mathcal{S} \to \mathbb{R}$ maps the extended history $(h, s)$ to a real number, i.e., the payment, made from the agent to the principal (but it can be negative). We remark that in principle, one can absorb payments into the action space. However, doing so would make the action space uncountable, introducing subtleties into the computational problem (which is the main focus of this paper). Here, we keep payments separate and explicit to avoid such issues. Also, our algorithm allows linear constraints on feasible payments, including but not limited to: nonnegative payments, no payments, etc. See Section 3.2 for more details.

**Utilities.** Fixing a mechanism $M = (\pi, p)$, we can then define the onward utility of the principal and the agent. Let $u_P^M : \mathcal{H} \times \mathcal{S} \to \mathbb{R}$ be the principal's onward utility function under mechanism $M$, defined inductively such that

$$u_P^M(h, s) = \sum_a \pi(h, s, a) \cdot \left( v_{|h|+1}^P(s, a) + \sum_{s'} P_{|h|+1}(s, a, s') \cdot u_P^M(h + (s, a), s') \right) + p(h, s),$$

with the boundary condition that $u_P^M(h, s) = 0$ for all $h \in \mathcal{H}_T$ and $s \in \mathcal{S}$. Here, all summations are over the entire state/action space. Let $u_P^M(\emptyset)$ be the overall utility of the principal, i.e.,

$$u_P^M(\emptyset) = \sum_s P_0(s) \cdot u_P^M(\emptyset, s).$$

Similarly, let $u_A^\pi : \mathcal{H} \times \mathcal{S} \to \mathbb{R}$ be the agent's onward utility function under mechanism $M$, defined such that

$$u_A^M(h, s) = \sum_a \pi(h, s, a) \cdot \left( v_{|h|+1}^A(s, a) + \sum_{s'} P_{|h|+1}(s, a, s') \cdot u_A^M(h + (s, a), s') \right) - p(h, s),$$

where $u_A^M(h, s) = 0$ for all $h \in \mathcal{H}_T$ and $s \in \mathcal{S}$. And let $u_A^M(\emptyset)$ be the overall utility of the agent, i.e.,

$$u_A^M(\emptyset) = \sum_s P_0(s) \cdot u_A^M(\emptyset, s).$$

We remark that while the above definition assumes that the principal cares about payments as much as the agent does, in fact, our algorithm allows for the principal to care about payments in an arbitrary linear way (including possibly not at all). See Section 3.2 for a detailed discussion.

**Incentive-compatible mechanisms.** We say a mechanism $M$ is incentive-compatible (IC) if the agent can never achieve a higher overall utility by misreporting the state, even in sophisticated ways. Formally, a reporting strategy $r : \mathcal{H} \times \mathcal{S} \to \mathcal{S}$ maps each history $h$ extended with the current state $s$ to a reported state $s'$, which is possibly different from $s$. Note that the agent only (mis)reports the current state, since the principal can memorize all historical reports. This reporting strategy induces a reported history $r(h) = (s_1', a_1, \dots, s_t', a_t)$ for each actual history $h = (s_1, a_1, \dots, s_t, a_t)$, where for each $i \in [t]$,

$$s_i' = r((s_1, a_1, \dots, s_{i-1}, a_{i-1}), s_i).$$

Note that we abuse notation here: in particular, $r(h, s)$ denotes a reported state, whereas $r(h)$ denotes a reported history. And without loss of generality, we only consider deterministic reporting strategies. Given a mechanism $M$ and a reporting strategy $r$, we can define the agent's utility function $u_A^{M,r}$ under $M$ and $r$ inductively such that

$$u_A^{M,r}(h, s) = \sum_a \pi(r(h), r(h, s), a) \cdot \left( v_{|h|+1}^A(s, a) + \sum_{s'} P_{|h|+1}(s, a, s') \cdot u_A^{M,r}(h + (s, a), s') \right)$$
$$- p(r(h), r(h, s)),$$

where $u_A^{M,r}(h, s) = 0$ for all $h \in \mathcal{H}_T$ and $s \in \mathcal{S}$. And let $u_A^{M,r}(\emptyset)$ be the overall utility of the agent, i.e.,

$$u_A^{M,r}(\emptyset) = \sum_s P_0(s) \cdot u_A^{M,r}(\emptyset, s).$$

In words, $u_A^{M,r}$ is the utility function of the agent applying the reporting strategy $r$ in response to the mechanism $M$. The mechanism $M$ is IC iff for any such reporting strategy $r$,

$$u_A^M(\emptyset) \geq u_A^{M,r}(\emptyset).$$

Since the revelation principle holds in dynamic environments (see, e.g., [26]), we focus on IC mechanisms in the rest of the paper.[1]

---

[1] Of course, the revelation principle will not hold in our dynamic setting if we allow it to generalize a static setting in which the revelation principle does not hold. For example, in the case of partial verification — not every type being able to misreport every other type — or costly misreporting, the revelation principle is known to hold only under certain conditions [23]. In this paper, we only consider the standard mechanism design setting in which every type can freely misreport any other type, but our techniques can be generalized to the other settings as well.

**Individually-rational mechanisms.** When payments are allowed, it is standard to impose individual-rationality (IR) (also known as voluntary-participation) constraints on the mechanism, which roughly say that the agent should never be made worse off by participating in the mechanism. In this paper, we consider two versions of IR constraints:

- A mechanism $M$ is overall IR if the overall utility of the agent is nonnegative, i.e., $u_A^M(\emptyset) \geq 0$. This ensures that the agent is willing to participate in the overall mechanism.

- A mechanism $M$ is dynamic IR if the onward utility of the agent is nonnegative for every history $h$ and current state $s$, i.e., $u_A^M(h, s) \geq 0$ for all $h \in \mathcal{H}$ and $s \in \mathcal{S}$. This stronger notion of IR further ensures that the agent has no incentive to leave the mechanism at any time.

As discussed in later sections, our algorithms work for all 3 cases regarding IR constraints: no IR (which results in an unbounded objective value if payments are allowed and valued by the principal), overall IR, and dynamic IR.

## 3 Computation of Optimal Mechanism

In this section, we investigate the computational problem of finding an optimal dynamic mechanism, which maximizes the principal's overall utility. For concreteness, we assume that all components of the dynamic environment, including the time horizon $T$, state and action spaces $\mathcal{S}$ and $\mathcal{A}$, valuation functions $v^P$ and $v^A$, and transition operator $P$, are given explicitly as input.

### 3.1 Hardness Result for Long-Horizon Environments

First we show that the problem with an arbitrarily large time horizon $T$ is intractable. In general, it takes exponentially many parameters in $T$ to describe a dynamic mechanism, which immediately rules out the possibility of computing a flat representation of an optimal mechanism in polynomial time. However, this leaves the possibility of computing succinct representations, e.g., an oracle which maps extended histories to distributions over actions. Our hardness result shows that it is hard to approximate the principal's maximum utility within a constant factor, which rules out the possibility of such succinct representations that can be efficiently evaluated, assuming $\mathsf{P} \neq \mathsf{NP}$. The proof of the theorem, as well as all other proofs, are deferred to the appendices.

**Theorem 1.** *When the time horizon $T$ can be arbitrarily large, it is* $\mathsf{NP}$*-hard to approximate the principal's maximum utility within a factor of* $7/8 + \varepsilon$ *for any* $\varepsilon > 0$.

### 3.2 Algorithm for Short-Horizon Environments

Now we give a polynomial-time algorithm for computing an optimal mechanism when $T$ is a constant. Our algorithm is based on a delicate linear program (LP) formulation, which relies on the following notation and concepts.

**Feasible history-state pairs.** A history-state pair $(h, s)$, where $h = (s_1, a_1, \ldots, s_t, a_t)$, is $i$-feasible if $P_j(s_j, a_j, s_{j+1}) > 0$ for every $j \in \{i, i+1, \ldots, t-1\}$, and $P_t(s_t, a_t, s) > 0$. In other words, starting from $s_i$ and taking the actions specified in $h$, there is a positive probability that the rest of the history and the state $s$ are generated from the transition operator. We say a pair $(h, s)$ is feasible if it is 1-feasible.

**Feasible extensions.** For two history-state pairs $(h, s)$ and $(h', s')$ where $h = (s_1, a_1, \ldots, s_t, a_t)$ and $h' = (s'_1, a'_1, \ldots, s'_{t'}, a'_{t'})$, we say that $(h', s')$ feasibly extends $(h, s)$, i.e., $(h, s) \subseteq (h', s')$, if $(h, s) = (h', s')$, or the following conditions hold simultaneously:

- $t = |h| < |h'| = t'$.
- For any $i \in [t]$, $(s_i, a_i) = (s'_i, a'_i)$ (this holds automatically when $h = \emptyset$ and therefore $|h| = 0$).
- $s = s'_{t+1}$.
- $(h', s')$ is $(|h| + 1)$-feasible (note that this does not require $h$ itself to be feasible).

$$\textbf{objective:} \quad \max \sum_{h\in\mathcal{H}, s\in\mathcal{S}:(h,s) \text{ is feasible}} \left( \sum_{a\in\mathcal{A}} v^P_{|h|+1}(s,a) \cdot x(h,s,a) + y(h,s) \right) \tag{1}$$

$$\textbf{flow constraints:} \quad z(h,s) = \sum_{a\in\mathcal{A}} x(h,s,a) \qquad\qquad \forall h\in\mathcal{H}, s\in\mathcal{S} \tag{2}$$

$$z(\emptyset,s) = P^E_0(s) \qquad\qquad\qquad \forall s\in\mathcal{S} \tag{3}$$

$$z(h+(s,a),s') = P^E_{|h|+1}(s,a,s') \cdot x(h,s,a) \quad \forall h\in\mathcal{H}, s,s'\in\mathcal{S}, a\in\mathcal{A} \tag{4}$$

$$\textbf{utility:} \quad u(h,s) = \sum_{h'\in\mathcal{H}, s'\in\mathcal{S}:(h,s)\subseteq(h',s')} \left( \sum_{a\in\mathcal{A}} v^A_{|h'|+1}(s',a) \cdot x(h',s',a) - y(h',s') \right) \quad \forall h\in\mathcal{H}, s\in\mathcal{S} \tag{5}$$

$$\textbf{IC constraints:} \quad u(h,s,s') = \sum_{a\in\mathcal{A}} v^A_{|h|+1}(s,a) \cdot x(h,s',a) - y(h,s')$$
$$+ \sum_{a\in\mathcal{A}, s''\in\mathcal{S}} \frac{P_{|h|+1}(s,a,s'')}{P^E_{|h|+1}(s',a,s'')} \cdot u(h+(s',a),s'') \quad \forall h\in\mathcal{H}, s,s'\in\mathcal{S} \tag{6}$$

$$u(h,s) \geq \frac{P^E_{|h|}(s_p,a_p,s)}{P^E_{|h|}(s_p,a_p,s')} \cdot u(h,s,s'), \text{where } (s_p,a_p) = \text{last}(h) \quad \forall h\in\mathcal{H}, s,s'\in\mathcal{S} \tag{7}$$

$$\textbf{IR constraints:} \quad u(h,s) \geq 0 \qquad\qquad\qquad \forall h\in\mathcal{H}, s\in\mathcal{S} \tag{8}$$

$$\textbf{feasible actions:} \quad x(h,s,a) \geq 0 \qquad\qquad\qquad \forall h\in\mathcal{H}, s\in\mathcal{S}, a\in\mathcal{A} \tag{9}$$

$$\textbf{feasible payments:} \quad y(h,s) \geq 0 \qquad\qquad\qquad \forall h\in\mathcal{H}, s\in\mathcal{S} \tag{10}$$

Figure 1: Linear program for computing an optimal dynamic mechanism.

**Extended transition operator.** For notational simplicity we define the following extended transition operator $P^E_t : \mathcal{S} \times \mathcal{A} \to \Delta(\mathcal{S})$ for all $t \in \{0\} \cup [T]$, such that

$$P^E_t(s,a,s') = \begin{cases} P_t(s,a,s'), & \text{if } P_t(s,a,s') > 0 \\ 1, & \text{otherwise} \end{cases}.$$

In words, the extended transition operator assigns phantom probability 1 to each way of transitioning that happens with probability 0 (so $P^E_t(s,a)$ does not always normalize to 1). As a shorthand, let $P^E_0(s') = P^E_0(s,a,s')$ for some $s \in \mathcal{S}$ and $a \in \mathcal{A}$ (the specific choice does not matter). The extended transition operator helps in constructing the flow and IC constraints below and simplifies the formulation. In particular, we always have $P^E_t(s,a,s') > 0$.

**Last state-action pair.** For a history $h \in \mathcal{H}$ where $h = (s_1, a_1, \ldots, s_t, a_t)$, we use $\text{last}(h)$ as a shorthand for the last state-action pair, i.e., $\text{last}(h) = (s_t, a_t)$. In particular, when $h = \emptyset$, $\text{last}(h)$ can be any state-action pair (the choice does not affect our results — it is merely a simplifying shorthand).

Now we are ready to describe the LP formulation. The complete formulation is given in Figure 1. The formulation is for nonnegative payments and dynamic IR constraints — we will discuss later

how the formulation can be modified to allow other types of constraints. Below, we describe each of its components.

**Variables, flow constraints, and the corresponding mechanism.**    There are 5 classes of variables in the LP:

- $x(h, s, a)$: the absolute, unconditional probability that the mechanism reaches state $s$ via history $h$, and takes action $a$.
- $y(h, s)$: the payment for history-state pair $(h, s)$, scaled by the probability that the mechanism reaches $s$ via $h$ (i.e., $z(h, s)$).
- $z(h, s)$: the probability that the mechanism reaches state $s$ via history $h$, which by definition satisfies
$$z(h, s) = \sum_{a \in \mathcal{A}} x(h, s, a).$$
- $u(h, s)$: the onward utility of the agent at state $s$ with history $h$ assuming truthful reporting, scaled by the probability that the mechanism reaches $s$ via $h$ (i.e., $z(h, s)$).
- $u(h, s, s')$: the onward utility of the agent at state $s$ with history $h$ if the agent misreports $s'$, assuming truthful reporting in the future, scaled by the probability that the mechanism reaches $s'$ via $h$ (i.e., $z(h, s')$).

The flow constraints (Eq. (2)-(4)) enforce roughly the above interpretation of variables to $x(h, s, a)$ and $z(h, s)$, except for ways of transition that have probability $0$. For each way of transition with probability $0$, the extended transition operator assigns phantom probability $1$. This phantom probability is not counted in the objective function (because only feasible history-state pairs are counted) or in the utility variables $u(h, s)$ (because only feasible extensions are counted). So, the phantom probability does not affect the principal's or the agent's utility assuming truthful reporting. Instead, together with other constraints, it guarantees that the mechanism behaves well even for history-state pairs that appear with probability $0$ under truthful reporting, which is necessary for the mechanism to be IC (see later paragraphs). Under the above interpretation, the LP variables (and in particular, $x(h, s, a)$, $y(h, s)$ and $z(h, s)$) naturally correspond to a mechanism $M = (\pi, p)$. Formally, for each $h \in \mathcal{H}$, $s \in \mathcal{S}$:

- If $z(h, s) > 0$, then
$$p(h, s) = y(h, s)/z(h, s),$$
and for each $a \in \mathcal{A}$,
$$\pi(h, s, a) = x(h, s, a)/z(h, s).$$
- If $z(h, s) = 0$, then let $\pi(h, s)$ be an arbitrary distribution over $\mathcal{A}$, and $p(h, s) = 0$.

The feasibility of the mechanism (i.e., every $\pi(h, s)$ is a distribution over $\mathcal{A}$ and every $p(h, s)$ is nonnegative) is guaranteed by constraints (2), (9) and (10). We remark that while the mechanism constructed from the LP variables may not be unique, effectively this makes no difference, since the parts of the mechanism that are chosen arbitrarily can never be accessed when executing the mechanism. This is because $z(h, s) = 0$ only if at some point in the history $h$, there is an action that the mechanism would never play given the reported states and actions before that. In particular, the above does not simply apply to all history-state pairs $(h, s)$ that are reached with probability $0$ under truthful reporting, in which case $z(h, s)$ may still be positive due to the extended transition operator. Moreover, given any mechanism, one can construct LP variables in a similar way, such that the mechanism constructed from these variables is the same as the original mechanism (modulo the unreachable parts). In other words, the above correspondence is effectively bijective.

**The objective.**    The objective function of the LP (Eq. (1)) is precisely the overall utility of the principal under the mechanism constructed above, assuming truthful reporting. This is captured by the following lemma.

**Lemma 1.** *Let $M = (\pi, p)$ be the mechanism constructed from variables $x(h, s, a)$, $y(h, s)$, and $z(h, s)$ which satisfy the flow constraints. Then*

$$u_P^M(\emptyset) = \sum_{h \in \mathcal{H}, s \in \mathcal{S}:(h,s) \text{ is feasible}} \left( \sum_{a \in \mathcal{A}} v_{|h|+1}^P(s, a) \cdot x(h, s, a) + y(h, s) \right).$$

From this lemma, it is clear that the objective of the LP is the natural quantity to maximize.

**Utility.** The utility constraints (Eq. (5)) collect the agent's onward utility, where $u(h, s)$ is equal to the agent's onward utility in state $s$ from history $h$, assuming truthful reporting, scaled by $z(h, s)$. This is captured by the following lemma.

**Lemma 2.** *Let $M = (\pi, p)$ be the mechanism constructed from variables $x(h, s, a)$, $y(h, s)$, and $z(h, s)$ which satisfy the flow and utility constraints. For all $h \in \mathcal{H}$, $s \in \mathcal{S}$,*

$$u(h, s) = z(h, s) \cdot u_A^M(h, s).$$

The proof of Lemma 2 is essentially the same as that of Lemma 1. Given the correspondence to the agent's utility $u_A^M(h, s)$, the utility variables $u(h, s)$ act as auxiliary variables in IC constraints.

**IC constraints.** IC constraints are a key component of the LP formulation. There are two families of IC constraints: collecting the agent's scaled utility from single-step misreporting (Eq. (6)), and subsequently restricting the mechanism such that there is no incentive for misreporting (Eq. (7)). In Eq. (6), we build variables $u(h, s, s')$, which is supposed to be the onward utility of the agent in state $s$ from history $h$ misreporting $s'$, assuming truthful reporting in the future, scaled by $z(h, s')$ (rather than $z(h, s)$). This is captured by the following lemma.

**Lemma 3.** *Let $M = (\pi, p)$ be the mechanism constructed from variables $x(h, s, a)$, $y(h, s)$, and $z(h, s)$ which satisfy the flow constraints, the utility constraints, and Eq. (6). Then the following statement holds: for all $h \in \mathcal{H}$, $s, s' \in \mathcal{S}$, let reporting strategy $r_{h,s,s'}$ be such that*

$$r_{h,s,s'}(h', s'') = \begin{cases} s', & \text{if } h = h' \text{ and } s = s'' \\ s'', & \text{otherwise} \end{cases}.$$

*That is, $r_{h,s,s'}$ misreports $s'$ only in state $s$ from history $h$, and reports truthfully otherwise. Then for all $h \in \mathcal{H}$, $s, s' \in \mathcal{S}$,*

$$u(h, s, s') = z(h, s') \cdot u_A^{M, r_{h,s,s'}}(h, s).$$

Given Lemma 3, Eq. (7) then guarantees that the mechanism $M$ is robust against single-step misreporting for all reachable history-state pairs.

**Lemma 4.** *Let $M = (\pi, p)$ be the mechanism constructed from variables $x(h, s, a)$, $y(h, s)$, and $z(h, s)$ which satisfy the flow constraints, the utility constraints, and Eq. (6). The following is true if and only if the LP variables also satisfy Eq. (7): for all $h \in \mathcal{H}$, $s, s' \in \mathcal{S}$ where $(h, s)$ is reachable by the mechanism $M$,*

$$u_A^M(h, s) \geq u_A^{M, r_{h,s,s'}}(h, s).$$

We then show that a mechanism is IC if and only if there is no incentive for single-step misreporting, which directly implies that the mechanism $M$ constructed from the LP variables is IC. This is captured by the following lemma.

**Lemma 5.** *Let $M = (\pi, p)$ be the mechanism constructed from variables $x(h, s, a)$, $y(h, s)$, and $z(h, s)$ which satisfy the flow constraints, the utility constraints, and Eq. (6). Then $M$ is IC if and only if the LP variables also satisfy Eq. (7).*

**IR constraints, feasible actions, and feasible payments.** These constraints are straightforward given the correspondence between the LP variables and the mechanism that we have discussed above. Note that while Eq. (8) is for dynamic IR (i.e., the agent has no incentive to leave the mechanism at any point) and Eq. (10) is for nonnegative payments, it is easy to replace them with similar constraints that correspond to overall IR or no payments. See Appendix C for more details.

**Optimality of LP solution.** Given the above facts, we are ready to state and prove the main result of the paper.

**Theorem 2.** *There is an algorithm which computes an optimal IC and (optionally) IR dynamic mechanism, with or without payments, in time $O(\mathrm{poly}(|\mathcal{S}|^T, |\mathcal{A}|^T, L))$, where $L$ is the number of bits required to encode each of the input parameters. In particular, when $T$ is constant, the algorithm runs in polynomial time.*

# 4 The Case of Myopic Agents: Characterization and Faster Algorithm

In this section, we briefly discuss a special case of the problem of computing optimal dynamic mechanisms, namely the case where the agent is myopic, or, equivalently, the agent has a discount factor of $0$. While our LP-based algorithm still applies, as we will see below, optimal mechanisms for myopic agents enjoy a succinct representation in this case, which also enables a faster algorithm that scales only linearly in the time horizon $T$. See Appendix D for more details, including the formal definition of myopic agents and the complete description of the algorithm.

## 4.1 Characterization of Optimal Mechanisms

We first show that when the agent is myopic, without loss of generality, the actions and payments specified by an optimal mechanism depend only on the time, the previous state, the previous action and the current state (we call such a mechanism a *succinct mechanism*), instead of the entire history-state pair.

**Lemma 6.** *Fix a dynamic environment. When the agent is myopic, for any IC mechanism $M = (\pi, p)$, there is another IC mechanism $M' = (\pi', p')$ (which is IR whenever $M$ is) such that*

- $u_P^{M'}(\emptyset) \geq u_P^M(\emptyset)$, and

- *for all $h \in \mathcal{H}$, $s \in \mathcal{S}$, $\pi'$ and $p'$ depend only on $|h|$, $s_p$, $a_p$ and $s$, where $(s_p, a_p) = \text{last}(h)$.*

*Moreover, the above is true regardless of whether payments are allowed, or which IR constraints are required.*

## 4.2 Faster Algorithm for Myopic Agents

Based on the above characterization, we present a faster algorithm for computing an optimal mechanism in the face of a myopic agent. In particular, the time complexity of this algorithm depends only linearly on the time horizon $T$, making it feasible for dynamic environments with a long time horizon. This is in contrast with the case of patient agents, for which, as we have seen, the long-horizon problem is hard to approximate. The algorithm uses as a subroutine a blackbox algorithm that computes an optimal IC (and optionally IR) mechanism in static environments, with or without payments. It is known that such an algorithm can be implemented using linear programming, and in some cases in more efficient ways [10, 12, 37].

**Theorem 3.** *When the agent is myopic, Algorithm **??** computes an optimal IC and (optionally) IR dynamic mechanism, with or without payments, in time*

$$O(T|\mathcal{S}||\mathcal{A}| \cdot T_{\text{stat}}(|\mathcal{S}|, |\mathcal{A}|, L)) = O(T \cdot \text{poly}(|\mathcal{S}|, |\mathcal{A}|, L)),$$

*where $T_{\text{stat}}$ is the time complexity of the blackbox algorithm used for computing an optimal IC (and optionally IR) mechanism in static environments, and $L$ is the number of bits required to encode each of the input parameters.*

# 5 Conclusion

We studied automated dynamic mechanism design and showed that, while it is computationally hard to find (even approximately) optimal mechanisms when (1) facing a patient agent and (2) the horizon is long, when either of these two conditions is dropped, an optimal mechanism can be found efficiently. An interesting future direction is to generalize our results to related problems with a stronger learning flavor, e.g., reinforcement learning with IC and/or IR constraints.

Besides using our algorithms directly for appropriate applications, the experimental results that they enable (including those that we presented in Appendix F) can guide new theory. For example, can we rigorously prove the benefit of facing a patient agent when the setting is not all too adversarial, and perhaps even characterize the transition point at which facing a patient agent becomes better than facing a myopic one? Analytically derived mechanisms can also be compared to these experimental results to see how close to optimal in performance they are. Finally, close inspection of the actual mechanisms generated by our algorithms may reveal insights that can be used to analytically design new mechanisms.

## Funding Transparency Statement

Funding in direct support of this work: NSF grant IIS-1814056.

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
