## A   Overview of Results in Appendices

In Section C, we discuss ways of customizing the LP formulation given in Section 3 to accommodate richer objectives and/or constraints, such as feasible intervals of payments, different IR constraints and discount factors. We also present an integer LP formulation for finding optimal deterministic mechanisms.

In Section D, we zoom into a special case of the problem where the agent is *myopic*, i.e., where the agent cares only about immediate value when making decisions. This is still practically meaningful, since it is commonly assumed and observed that the principal is often much more patient than the agent in dynamic environments. (This could also correspond to the agent really being a sequence of short-lived agents; for example, there may be high turnover in the research group in the example above, where each researcher is there only for one period.) We show that in such cases, without loss of generality, optimal mechanisms admit succinct representations, i.e., they depend only on the current state and time, the previous state, and the previous action. Based on this characterization, we provide an improved algorithm for finding optimal mechanisms in the face of a myopic agent, whose time complexity depends *linearly* on the time horizon. As a result, this algorithm scales well in

dynamic environments with long time horizons, which is in sharp contrast to the general case where long time horizons lead to inapproximability.

As discussed above, without strategic behavior, our problem degenerates to the problem of planning in (finite episodic) Markov Decision Processes (MDPs). It is known that in MDPs, optimal strategies are without loss of generality *memoryless*: they depend only on the current time and state. To this end, one may wonder if memoryless mechanisms are also (approximately) optimal and/or easier to compute in dynamic environments with strategic behavior. In Section E, we give a negative answer to the above question, by showing that (1) the principal's optimal utility achieved by memoryless mechanisms can be arbitrarily worse than that achieved by general dynamic mechanisms, and (2) it is NP-hard to approximate the principal's optimal utility achieved by memoryless mechanisms within a factor of $(7/8 + \varepsilon)$ for any $\varepsilon > 0$. In other words, memoryless mechanisms do not provide a good solution for our problem, in terms of both optimality and computational tractability.

Finally, in Section F, we apply our algorithms to synthetic dynamic environments with different characteristics, in order to provide a proof of concept for the methods we propose, as well as to explore various phenomena in dynamic environments and their implications for (automated) dynamic mechanism design. Below are some of our key findings:

- As in static environments, taking into consideration the agent's incentives in dynamic environments can greatly improve the principal's utility.

- In dynamic environments, optimal mechanisms are remarkably robust to misaligned interests between the principal and the agent, whereas the performance of naïve mechanisms (which disregard the agent's incentives) degrades much faster.

- Even when the principal's and the agent's valuations are perfectly aligned, an agent acting myopically can still considerably hurt the principal's utility in naïve mechanisms, but this can be largely corrected by using mechanisms that are optimal in the face of a myopic agent.

- As one would expect, patient agents are easier to cooperate with, and myopic agents are easier to exploit; however, even when the principal's and the agent's valuations are negatively correlated, it is possible to find a middle ground where cooperation is more beneficial than exploitation in the long run.

## B  Additional Related Work

**Repeated allocation without money.**  Another related line of work is devoted to studying the design of repeated allocation mechanisms without money, motivated for example by allocating shared computing resources over time [30, 27, 5, 29]. When there is no money, repeated allocation allows one to better take current preferences for the items into account, because one can "pay" for one's current allocation with one's future allocations. Indeed, a common theme of this line of work is to introduce artificial currencies or to approximate mechanisms *with* money via the use of future allocations. The algorithms we present here can be used to find optimal mechanisms without money directly.

**Equilibrium computation.**  Our main result can be viewed as an efficient algorithm for computing Stackelberg equilibria in a special class of extensive-form games. Equilibrium computation is quite well understood in normal-form games, where there are polynomial-time algorithms for computing a Stackelberg equilibrium [19], or a Nash equilibrium when the game is zero-sum (see, e.g., [54]), in two-player games, whereas finding a Nash equilibrium in general-sum two-player games is already PPAD-complete [21, 16]. For extensive-form games, von Stengel [55] and Koller et al. [36] propose the sequence-form representation, which leads to an efficient algorithm for finding a Nash equilibrium (which is also a Stackelberg equilibrium) in two-player zero-sum games. However, as shown by Letchford and Conitzer [38], computing a Stackelberg equilibrium in two-player general-sum extensive-form games is NP-hard in general. Polynomial-time (exact or $(1 - \varepsilon)$-approximation) algorithms are known only for highly restrictive cases, e.g., in perfect-information settings [11], or when the follower is a finite state machine with limited memory [15] (although practically scalable algorithms exist for more general settings [10, 13, 14, 37]). Our results push the boundary of tractability of Stackelberg equilibrium in extensive-form games, by enabling efficient computation in a nontrivial class of *general-sum* extensive-form games with *imperfect information*.

## C Customizing the LP Formulation.

The LP formulation in Figure 1 allows for nonnegative payments, assumes that the principal cares about payments as much as the agent, and enforces dynamic IR constraints. As mentioned above, one can customize all these components by modifying the corresponding parts of the LP formulation. Below we discuss several ways of customization.

- **Unequal valuations for payments**: in the case where the principal has utility $c$ for one unit of payment (whereas without loss of generality the agent has utility 1), one may replace the objective function (Eq. (1)) with

$$\max \sum_{h \in \mathcal{H}, s \in \mathcal{S}:(h,s) \text{ is feasible}} \left( \sum_{a \in \mathcal{A}} v^P_{|h|+1}(s,a) \cdot x(h,s,a) + c \cdot y(h,s) \right).$$

  Note that our formulation works only when the principal cares linearly about payments. Notably, the principal may not care about payments at all (as in the case of paying the agent in "brownie points"), or even dislike payments made by the agent (as in the case where the agent is asked to expend useless effort or "burn money" and the principal cares in part about the resulting loss of welfare).

- **No payments**: to forbid payments in the mechanism, one can simply replace Eq. (10) with

$$y(h,s) = 0, \ \forall h \in \mathcal{H}, s \in \mathcal{S}.$$

- **Feasible intervals of payments**: more generally, one may wish to specify a feasible interval $[a_{h,s}, b_{h,s}]$ for the payment at each history-state pair $(h, s)$ such that $a_{h,s} \leq p(h, s) \leq b_{h,s}$, which subsumes both nonnegative payments and no payments as special cases. This can be done by replacing Eq. (10) with

$$a_{h,s} \cdot z(h,s) \leq y(h,s) \leq b_{h,s} \cdot z(h,s), \ \forall h \in \mathcal{H}, s \in \mathcal{S}.$$

- **Overall/no IR**: when the agent can choose whether to participate in the mechanism, but cannot leave halfway (corresponding to an overall IR constraint), one can replace Eq. (8) with

$$\sum_{s \in \mathcal{S}} u(\emptyset, s) \geq 0.$$

  Also, when leaving the mechanism is not an option for the agent from the very beginning (corresponding to no IR constraint), one may remove IR constraints simply by removing Eq. (8).

- **Discount factors**: to accommodate the case where the agent has a discount factor $0 \leq \delta < 1$, one can modify the LP formulation in the following way:
  - Replace Eq. (5) with

$$u(h,s) = \sum_{h' \in \mathcal{H}, s' \in \mathcal{S}:(h,s) \subseteq (h',s')} \delta^{|h'|-|h|} \cdot \left( \sum_{a \in \mathcal{A}} v^A_{|h'|+1}(s',a) \cdot x(h',s',a) - y(h',s') \right), \ \forall h \in \mathcal{H}, s \in \mathcal{S}.$$

  - Replace Eq. (6) with

$$u(h,s,s') = \sum_{a \in \mathcal{A}} v^A_{|h|+1}(s,a) \cdot x(h,s',a) - y(h,s')$$
$$+ \delta \cdot \sum_{a \in \mathcal{A}, s'' \in \mathcal{S}} \frac{P_{|h|+1}(s,a,s'')}{P^E_{|h|+1}(s',a,s'')} \cdot u(h+(s',a),s''), \ \forall h \in \mathcal{H}, s, s' \in \mathcal{S}$$

- **Deterministic mechanisms**: the problem of computing an optimal deterministic mechanism is NP-hard even in static environments [17, 18]. Nevertheless, given our LP formulation, one can restrict the mechanism to be deterministic by introducing Boolean variables, resulting in a mixed integer LP. While integer LPs are hard to solve in a worst-case sense, real-world problems often admit certain structures which can be exploited by commercial solvers such as CPLEX and Gurobi. To be specific, we introduce a Boolean variable $c(h,s,a)$ which controls $x(h,s,a)$

for all $h \in \mathcal{H}$, $s \in \mathcal{S}$, and $a \in \mathcal{A}$, and ensures that fixing $h$ and $s$, $x(h, s, a)$ can be positive for at most one action $a \in \mathcal{A}$. This is implemented by the following constraints (in addition to the existing ones):

$$x(h, s, a) \leq c(h, s, a) \qquad \forall h \in \mathcal{H}, s \in \mathcal{S}, a \in \mathcal{A}$$

$$\sum_{a \in \mathcal{A}} c(h, s, a) = 1 \qquad \forall h \in \mathcal{H}, s \in \mathcal{S}$$

$$c(h, s, a) \in \{0, 1\} \qquad \forall h \in \mathcal{H}, s \in \mathcal{S}, a \in \mathcal{A}.$$

We also remark that the above discussion is non-exhaustive: one can impose richer restrictions by modifying the LP formulation in other linear ways, and/or combining the above modifications.

## D  The Case of Myopic Agents: Characterization and Faster Algorithm

In this section, we consider a special case of the problem of computing optimal dynamic mechanisms, namely the case where the agent is myopic, or, equivalently, the agent has a discount factor of $0$. While our LP-based algorithm still applies, as we will see below, optimal mechanisms for myopic agents enjoy a succinct representation in this case, which also enables a faster algorithm that scales only linearly in the time horizon $T$.

**Myopic agents.**  The utility $u_A^M$ of a myopic agent under mechanism $M$ is such that

$$u_A^M(h, s) = \sum_a \pi(h, s, a) \cdot v_{|h|+1}^A(s, a) - p(h, s),$$

where $u_A^M(h, s) = 0$ for all $h \in \mathcal{H}_T$ and $s \in \mathcal{S}$. Given a reporting strategy $r$, the utility $u_A^{M,r}$ of the agent under mechanism $M$ and reporting strategy $r$ is

$$u_A^{M,r}(h, s) = \sum_a \pi(r(h), r(h, s), a) \cdot v_{|h|+1}^A(s, a) - p(r(h), r(h, s)).$$

$M$ is IC if and only if for all $h \in \mathcal{H}$ and $s \in \mathcal{S}$, there are no future reporting strategies that lead to better utility, i.e., for every reporting strategy $r$ where $r(h', s') = s'$ whenever $|h'| < |h|$,

$$u_A^M(h, s) \geq u_A^{M,r}(h, s).$$

Note that since the agent is myopic, it is insufficient to simply require $u_A^M(\emptyset) \geq u_A^{M,r}(\emptyset)$. Also, it is necessary to restrict misreporting to the future, since otherwise the agent would be allowed and sometimes incentivized to change the past, leading to unrealistically strong IC requirements. Again, since the revelation principle holds, we focus only on IC mechanisms.

### D.1  Characterization of Optimal Mechanisms

We first show that when the agent is myopic, without loss of generality, the actions and payments specified by an optimal mechanism depend only on the time, the previous state, the previous action and the current state (we call such a mechanism a *succinct mechanism*), instead of the entire history-state pair.

**Lemma 7.**  *Fix a dynamic environment. When the agent is myopic, for any IC mechanism $M = (\pi, p)$, there is another IC mechanism $M' = (\pi', p')$ (which is IR whenever $M$ is) such that*

- $u_P^{M'}(\emptyset) \geq u_P^M(\emptyset)$, *and*

- *for all $h \in \mathcal{H}$, $s \in \mathcal{S}$, $\pi'$ and $p'$ depend only on $|h|$, $s_p$, $a_p$ and $s$, where $(s_p, a_p) = \text{last}(h)$.*

*Moreover, the above is true regardless of whether payments are allowed, or which IR constraints are required.*

**Algorithm 1:** Algorithm for computing an optimal mechanism against a myopic agent.

---

**Input:** Time horizon $T$, transition probabilities $\{P_t\}_{t\in[T]}$, principal's valuation functions
$\{v_t^P\}_{t\in[T]}$, agent's valuation functions $\{v_t^A\}_{t\in[T]}$.

**Output:** An optimal IC (for a myopic agent) mechanism $M = (\pi, p)$.

**1 for** $t = T, T-1, \ldots, 1$ **do**

**2**     **for** $s \in \mathcal{S}$, $a \in \mathcal{A}$ **do**

**3**         let $u(t,s,a) \leftarrow v_t^P(s,a) + \sum_{s'\in\mathcal{S}} P_t(s,a,s') \cdot u_P^M(t+1,s,a,s')$;

        `/* the above operation is well-defined, in particular because`
            `` $u_P^M(t+1,s,a,s')$ `depends only on the part of` $M$ `that has already`
            `been computed` `*/`

**4**     **end**

**5**     **for** $s_p \in \mathcal{S}$, $a_p \in \mathcal{A}$ **do**

**6**         let $(\pi', p') \leftarrow \mathsf{OptStatMech}(\mathcal{S}, \mathcal{A}, \{P_{t-1}(s_p,a_p,s)\}_s, \{u(t,s,a)\}_{s,a}, \{v_t^A(s,a)\}_{s,a})$;

        `/* call OptStatMech to compute an optimal static mechanism` $(\pi', p')$`,`
            `in a static environment with type space` $\mathcal{S}$`, action space` $\mathcal{A}$`,`
            `population distribution` $\{P_{t-1}(s_p,a_p,s)\}_s$`, principal's utility`
            `function` $\{u(t,s,a)\}_{s,a}$`, and agent's utility function` $\{v_t^A(s,a)\}_{s,a}$
            `*/`

**7**         **for** $s \in \mathcal{S}$ **do**

**8**             let $\pi(t, s_p, a_p, s) \leftarrow \pi'(s)$, and $p(t, s_p, a_p, s) \leftarrow p'(s)$;

**9**         **end**

**10**     **end**

**11 end**

**12 return** $M = (\pi, p)$;

---

## D.2   Faster Algorithm for Myopic Agents

Based on the above characterization, we present below a faster algorithm for computing an optimal mechanism in the face of a myopic agent. In particular, the time complexity of this algorithm depends only linearly on the time horizon $T$, making it feasible for dynamic environments with a long time horizon. This is in contrast with the case of patient agents, for which, as we have seen, the long-horizon problem is hard to approximate.

To improve readability, we use the following shorthand notation for succinct mechanisms. For a succinct mechanism $M = (\pi, p)$, for any $h \in \mathcal{H}$ and $s \in \mathcal{S}$, let $\pi(t, s_p, a_p, s) = \pi(h, s)$ be the action policy at $(h, s)$, and $p(t, s_p, a_p, s) = p(h, s)$ be the payment function, where $(s_p, a_p) = \mathrm{last}(h)$ and $t = |h| + 1$. Also, observe that the principal's onward utility at any history-state pair $(h, s)$ depends only on the previous state $s_p$, the previous action $a_p$, and the current state $s$. In such cases, we also denote this utility by $u_P^M(t, s_p, a_p, s) = u_P^M(h, s)$.

The full algorithm is given as Algorithm 1. It uses as a subroutine an algorithm $\mathsf{OptStatMech}$ which computes an optimal IC (and optionally IR) mechanism in static environments, with or without payments. It is known that such an algorithm can be implemented using linear programming, and in some cases in more efficient ways [17, 19, 56]. Algorithm 1 proceeds in an inductive fashion, building a succinct mechanism backwards, one layer at a time. It repeatedly solves the problem of maximizing the principal's expected onward utility over the current state $s$, given the previous state $s_p$ and the previous action $a_p$. Since $s_p$ and $a_p$ together induce a roll-in distribution over the state space, this problem can be reduced to computing an optimal static mechanism, where the valuation function of the principal depends on the optimal mechanism in the following layers. This can then be solved by calling $\mathsf{OptStatMech}$, the algorithm for computing an optimal static mechanism. Below we state and prove the correctness and computational efficiency of Algorithm 1.

**Theorem 4.** *When the agent is myopic, Algorithm 1 computes an optimal IC and (optionally) IR dynamic mechanism, with or without payments, in time*

$$O(T|\mathcal{S}||\mathcal{A}| \cdot T_{\mathrm{stat}}(|\mathcal{S}|, |\mathcal{A}|, L)) = O(T \cdot \mathrm{poly}(|\mathcal{S}|, |\mathcal{A}|, L)),$$

*where $T_{\mathrm{stat}}$ is the time complexity of $\mathsf{OptStatMech}$, and $L$ is the number of bits required to encode each of the input parameters.*

**Customizing Algorithm 1.** We remark that Algorithm 1 can also be customized to allow for unequal valuations of payments, feasible intervals of payments, etc. Moreover, it can be adapted to compute an optimal deterministic mechanism, by requiring OptStatMech to compute an optimal deterministic static mechanism. Again, while this is generally hard to compute, for practical purposes, it is reasonable to expect that OptStatMech implemented using commercial mixed integer LP solvers (or in other practically efficient ways) can find an optimal mechanism efficiently.

# E    Infeasibility of Memoryless Mechanisms

From a planning perspective, automated dynamic mechanism design can be viewed equivalently as planning in MDPs where the current state cannot be directly observed, but instead, has to be reported by a strategic agent whose interest may not align with the planner's. In particular, when the planner and the agent share the same valuation function, automated dynamic mechanism design degenerates to the classical problem of planning in episodic MDPs with a finite planning horizon. In the latter problem, it is well known that without loss of generality, any optimal policy depends only on the time and the current state, i.e., it is memoryless. And moreover, such optimal policies can be computed in polynomial time. In light of the above facts, the following questions arise naturally: are there (approximately) optimal mechanisms that are also memoryless, and can we find optimal memoryless mechanisms efficiently? In this section, we give negative answers to both questions, which means memoryless mechanisms are generally infeasible for dynamic environments. We first show that memoryless mechanisms can be arbitrarily worse than general, history-dependent mechanisms, against both patient and myopic agents.

**Theorem 5.** *Regardless of whether the agent is myopic, for any $\varepsilon > 0$, there is a dynamic environment where the principal's utility under an optimal memoryless mechanism is at most an $\varepsilon$ fraction of the principal's optimal utility.*

Now we show that on top of the suboptimality, optimal memoryless mechanisms are computationally hard to approximate.

**Theorem 6.** *Regardless of whether the agent is myopic, it is NP-hard to approximate the principal's maximum utility under memoryless mechanisms within a factor of $7/8 + \varepsilon$ for any $\varepsilon > 0$.*

# F    Experimental Results

In this section, we present experimental results where our algorithms are applied to synthetic dynamic environments of different characteristics. The main goals of the experiments are

- to provide a proof of concept for the methods proposed in this paper,

- to illustrate the necessity of considering incentives when planning in dynamic environments (as opposed to disregarding the agent's valuations and treating the problem simply as an MDP based on the principal's valuations),

- to study the effect of cooperation and competition in dynamic mechanism design, and

- to understand the difference between patient and myopic agents from the principal's perspective, especially when the parameters of the environment vary.

## F.1    Setup of Experiments

**Mechanisms/models of the agent under consideration.** For each dynamic environment examined, we consider the following quantities from different combinations of mechanisms and models of the agent:

- **Naïve mechanisms facing a naïve agent**: the principal's optimal utility facing a naïve agent who always reports truthfully, i.e., the optimal utility when treating the problem simply as an MDP based on the principal's valuations.

- **Naïve mechanisms facing a patient agent**: the principal's utility, when executing the optimal mechanism/policy for naïve agents, facing a strategic agent who is patient.

- **Naïve mechanisms facing a myopic agent**: the principal's utility, when executing the optimal mechanism/policy for naïve agents, facing a strategic agent who is myopic.
- **Patient mechanisms facing a patient agent**: the principal's optimal utility facing a strategic agent who is patient.
- **Myopic mechanisms facing a myopic agent**: the principal's optimal utility facing a strategic agent who is myopic.

For simplicity, payments are not allowed in any of our experiments.

**Dynamic environments.** To manifest the effect of cooperation and competition, we generate synthetic dynamic environments in the following way:

- Fix the time horizon $T$, number of states $|\mathcal{S}|$, number of actions $|\mathcal{A}|$, and correlation parameter $\eta \in [-1, 1]$ (explained below).
- Let the initial distribution $P_0$ be a random distribution generated in the following way: for each state $s$, we generate a uniformly random real number $\mathrm{rand}(s)$ between 0 and 1, which is proportional to $P_0(s)$. That is, $P_0(s) = \mathrm{rand}(s)/\left(\sum_{s'} \mathrm{rand}(s')\right)$.
- For each $t \in [T]$, $s \in \mathcal{S}$ and $a \in \mathcal{A}$, we generate the transition distribution $P_t(s, a)$ independently in the same way that $P_0$ is generated.
- For each $t \in [T]$, $s \in \mathcal{S}$ and $a \in \mathcal{A}$, let $v_t^P(s, a)$ be an independent, uniformly random real number between 0 and 1.
- For each $t \in [T]$, $s \in \mathcal{S}$ and $a \in \mathcal{A}$, let $v_t^A(s, a) = \eta \cdot v_t^P(s, a) + (1 - |\eta|) \cdot \mathrm{rand}(t, s, a)$, where $\mathrm{rand}(t, s, a)$ is an independent, uniformly random real number between 0 and 1.

The correlation parameter $\eta$ controls the extent to which the interests of the principal and the agent are (mis)aligned. In particular, if $\eta = 1$, then the principal and the agent have exactly the same valuations, corresponding to full cooperation. If $\eta = -1$, then the principal and the agent are in a zero-sum situation, corresponding to full competition.

### F.2 Summary of Experimental Results

**Suboptimality of naïve mechanisms.** As we can see from Figure 2, even when the state and action spaces are extremely simple, i.e., there are only 2 states and 2 actions, when the correlation parameter $\eta = -1$ (i.e., when the agent acts adversarially), naïve mechanisms facing a strategic agent can only achieve about 75% of the naïve benchmark, i.e., the optimal utility when the agent is naïve. When $\eta = 0$ (i.e., when the agent's and principal's valuations are independent), naïve mechanisms facing a strategic agent still achieve only 85% of the naïve benchmark. On the other hand, the respective optimal mechanisms facing a patient or myopic agent consistently achieve about 95% of the naïve benchmark. This gap is further amplified in Figure 3: as the environment becomes more and more complex (i.e., the numbers of states and actions become larger and larger), the utility of naïve mechanisms facing a strategic agent drops below 20% of the naïve benchmark when $\eta = -1$, and to about 50% when $\eta = 0$. In contrast, the respective optimal mechanisms facing a patient or myopic agent still achieve about 70% of the naïve benchmark even when $\eta = -1$. These phenomena suggest that when the agent is not fully cooperative, taking strategic behavior into consideration significantly improves the principal's utility, even in extremely simple dynamic environments. Moreover, the more complex the environment is, the larger this gap becomes.

Another interesting fact to note is that even when the principal's and the agent's valuations are exactly the same (i.e., when $\eta = 1$), naïve mechanisms are still suboptimal facing a myopic agent, since the agent may sacrifice greater long-term gain in exchange for smaller immediate value. This phenomenon is more significant in Figure 2, especially in environments with longer time horizons. In such cases, taking into consideration the fact that the agent is myopic mitigates the loss, and recovers almost all the utility of the naïve benchmark.

**Effect of cooperation and competition.** As the correlation parameter increases, both Figure 2 and Figure 3 show clear upward trends in all the quantities that we consider (except for the naïve benchmark which is always normalized to 1), as one would expect. Nevertheless, we note the following facts from the figures: compared to naïve mechanisms, optimal mechanisms facing a

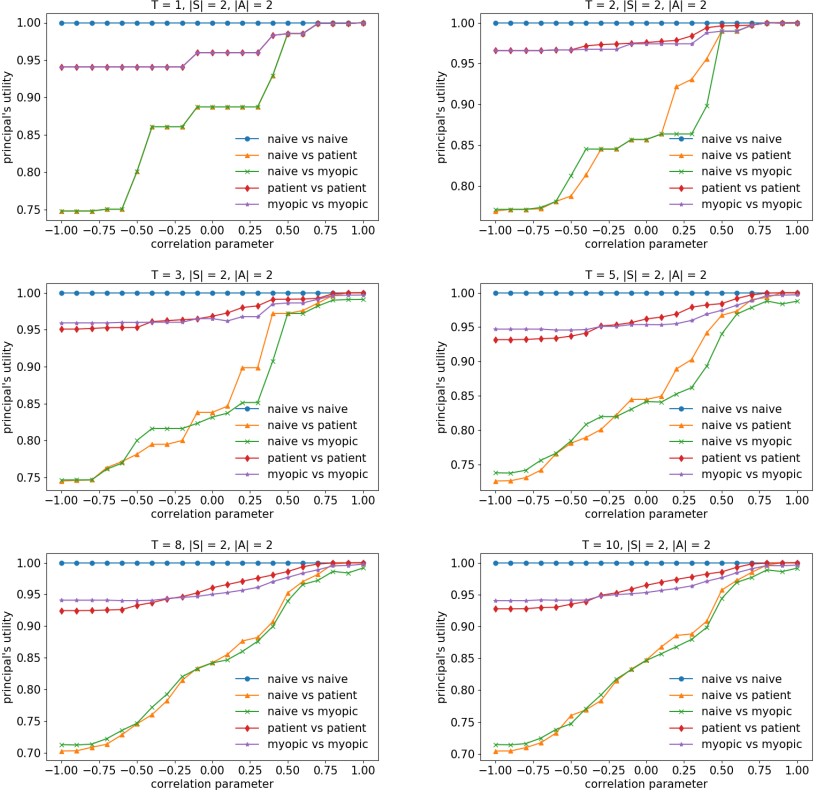

Figure 2: Performance of different mechanisms facing different types of agents when $|\mathcal{S}| = |\mathcal{A}| = 2$ and the time horizon $T$ varies. All numbers are normalized by the optimal utility facing a naïve agent. Every point is an average of $10$ independent runs using different random seeds.

strategic agent are much less affected by the correlation parameter. Moreover, as Figure 2 shows, the performance of optimal mechanisms facing a strategic agent is remarkably stable as the time horizon grows. In other words, in random dynamic environments, the utility loss caused by competing interests of the principal and the agent is only mildly amplified by long time horizons.

**Difference between patient and myopic agents.** As can be seen from the figures, regardless of whether the agent is patient or myopic, the principal's optimal utility is almost the same. Nevertheless, the difference appears to be amplified as the time horizon grows (see Figure 2). When the correlation parameter $\eta = -1$, the optimal utility facing a myopic agent is noticeably larger than that facing a patient agent — which makes sense as only the patient agent has interests truly opposite those of the principal. This gap shrinks as $\eta$ becomes larger, and vanishes when $\eta$ is around $-0.25$. Then, as $\eta$ continues to grow, the optimal utility facing a myopic agent falls behind and never catches up. In particular, when $\eta = 1$, the optimal utility facing a patient agent is the same as the naïve benchmark, whereas that facing a myopic agent is slightly smaller. The above phenomena indicate that in environments with a long time horizon, myopic agents are easier to exploit, while patient agents are easier to cooperate with. Interestingly, the critical value of $\eta$, where the optimal utility facing a patient agent catches up, is about $-0.25$ instead of $0$, which suggests that even when the principal's and the agent's valuations are mildly negatively correlated, it is possible to find a middle ground where cooperation is more beneficial than exploitation in the long run.

# G  Omitted Proofs from Section 3

*Proof of Theorem 1.* We consider the case where payments are not allowed, i.e., $p_t(h, s) = 0$ for all $h \in \mathcal{H}$ and $s \in \mathcal{S}$. The case with payments and dynamic IR constraints is essentially the same. We use a similar reduction from MAX-SAT to the ones in [45, 40] for partially observable

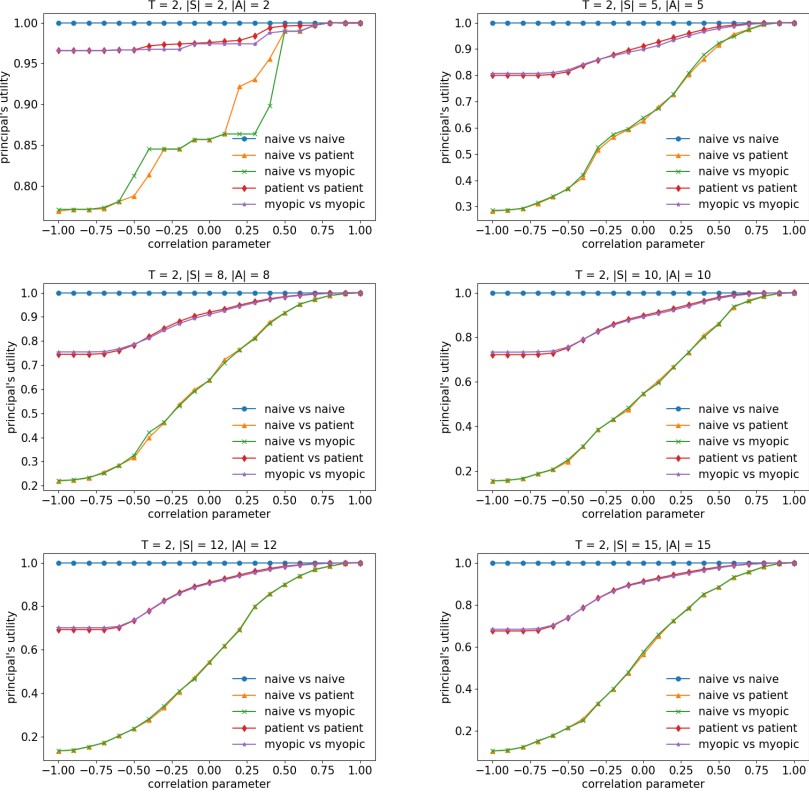

Figure 3: Performance of different mechanisms facing different types of agents when $T = 2$ and the numbers of states and actions, $|\mathcal{S}|$ and $|\mathcal{A}|$, vary. All numbers are normalized by the optimal utility facing a naïve agent. Every point is an average of $10$ independent runs using different random seeds.

Markov decision processes (POMDPs). Given a MAX-SAT instance with $n$ variables $x_1, \dots, x_n$ and $m$ clauses $c_1, \dots, c_m$ where $c_i = \{\ell_{i,j}\}_{j \in [k_i]}$ and each $\ell_{i,j}$ is a literal, we construct a dynamic environment where $T = n$, $|\mathcal{S}| = m + 1$, and $|\mathcal{A}| = 2$. The goal is to show that the maximum utility is precisely the fraction of clauses that can be simultaneously satisfied. Without loss of generality, we assume that no clause contains both the positive literal and the negative literal of a same variable. We first describe $\mathcal{S}$ and $\mathcal{A}$. Each clause $c_i$ corresponds to a unique state in $\mathcal{S}$, $s_i$. In addition to these $m$ states, there is another state $s_0$. $\mathcal{A}$ consists of two actions: $a_{\text{pos}}$ and $a_{\text{neg}}$. The transition operator and the principal's valuation function are such that:

- The initial distribution is uniform over $\{s_i\}_{i \in [m]}$, i.e., $P_0(s_i) = 1/m$ for each $i \in [m]$.

- For each $t \in [T]$ and $a \in \mathcal{A}$,

$$P_t(s_0, a, s_0) = 1 \quad \text{and} \quad v_t^P(s_0, a) = 0.$$

Moreover, for each $t \in [T]$ and $i \in [m]$:

- If $x_t^+ \in c_i$, then

$$P_t(s_i, a_{\text{pos}}, s_0) = P_t(s_i, a_{\text{neg}}, s_i) = 1,$$

and

$$v_t^P(s_i, a_{\text{pos}}) = 1 \quad \text{and} \quad v_t^P(s_i, a_{\text{neg}}) = 0.$$

- if $x_t^- \in c_i$, then

$$P_t(s_i, a_{\text{pos}}, s_i) = P_t(s_i, a_{\text{neg}}, s_0) = 1,$$

and

$$v_t^P(s_i, a_{\text{pos}}) = 0 \quad \text{and} \quad v_t^P(s_i, a_{\text{neg}}) = 1.$$

– otherwise,

$$P_t(s_i, a_{\mathrm{pos}}, s_i) = P_t(s_i, a_{\mathrm{neg}}, s_i) = 1,$$

and

$$v_t^P(s_i, a_{\mathrm{pos}}) = v_t^P(s_i, a_{\mathrm{neg}}) = 0.$$

- The principal and the agent are in a zero-sum situation, i.e., for any $t \in [T]$, $s \in \mathcal{S}$, $a \in \mathcal{A}$,

$$v_t^A(s, a) = 1 - v_t^P(s, a).$$

Now we show that the maximum utility is precisely the fraction of clauses that can be simultaneously satisfied. First observe that without loss of generality, an optimal mechanism depends only on time (and not on the reported states). This is because of the zero-sum situation: if the mechanism depends on the reports, then the agent can always choose the worst sequence of actions, which can only make the principal's utility smaller. Moreover, given the above observation, without loss of generality, an optimal mechanism is deterministic. This is because the overall utility of the principal is linear in the action at any time $t$, so one can always round a randomized mechanism into a deterministic one with at least the same overall utility.

Given the above observations, an optimal mechanism corresponds precisely to a way of assigning values to variables in the MAX-SAT instance: for each $t \in [T]$, the action at time $t$ is $a_{\mathrm{pos}}$ iff the variable $x_t = 1$ (i.e., the literal $x_t^+$ is chosen). Moreover, when the initial state is $s_i$, the onward utility is 1 if the clause $c_i$ is satisfied by the above assignment, and 0 otherwise. Since the initial state is uniformly at random among $\{s_i\}_{i \in [m]}$, the maximum utility is precisely the maximum fraction of clauses that are satisfiable by some assignment. The theorem then follows from the fact that MAX-SAT is hard to approximate within a factor of $7/8 + \varepsilon$ for any $\varepsilon > 0$ [32]. $\qquad\square$

*Proof of Lemma 1.* For brevity, let $\mathrm{obj}$ denote the objective, i.e.,

$$\mathrm{obj} = \sum_{h \in \mathcal{H}, s \in \mathcal{S}:(h,s) \text{ is feasible}} \left( \sum_{a \in \mathcal{A}} v_{|h|+1}^P(s, a) \cdot x(h, s, a) + y(h, s) \right).$$

Moreover, for each $h \in \mathcal{H}$, $s \in \mathcal{S}$, let

$$\mathrm{obj}(h, s) = \sum_{h' \in \mathcal{H}, s' \in \mathcal{S}:(h,s) \subseteq (h',s')} \left( \sum_{a \in \mathcal{A}} v_{|h'|+1}^P(s', a) \cdot x(h', s', a) + y(h', s') \right).$$

Observe that

$$\mathrm{obj} = \sum_{s \in \mathcal{S}} \mathrm{obj}(\emptyset, s).$$

We first prove inductively that for each $h \in \mathcal{H}$, $s \in \mathcal{S}$,

$$\mathrm{obj}(h, s) = z(h, s) \cdot u_P^M(h, s).$$

When $|h| = T - 1$, by the definition of feasible extensions and the construction of the mechanism,

$$\begin{aligned}
\mathrm{obj}(h, s) &= \sum_{a \in \mathcal{A}} v_T^P(s, a) \cdot x(h, s, a) + y(h, s) \\
&= z(h, s) \cdot \left( \sum_{a \in \mathcal{A}} v_T^P(s, a) \cdot \pi(h, s, a) + p(h, s) \right) \\
&= z(h, s) \cdot u_P^M(h, s).
\end{aligned}$$

When $|h| < T - 1$, for similar reasons,

$$\text{obj}(h, s) = \sum_{h',s':(h,s)\subseteq(h',s')} \left( \sum_{a\in\mathcal{A}} v^P_{|h'|+1}(s', a) \cdot x(h', s', a) + y(h', s') \right)$$

$$= \sum_{a\in\mathcal{A}} v^P_{|h|+1}(s, a) \cdot x(h, s, a) + y(h, s)$$

$$+ \sum_{h',s':(h,s)\subseteq(h',s'),|h'|>|h|} \left( \sum_{a\in\mathcal{A}} v^P_{|h'|+1}(s', a) \cdot x(h', s', a) + y(h', s') \right)$$

$$= z(h, s) \cdot \left( \sum_{a\in\mathcal{A}} v^P_{|h|+1}(s, a) \cdot \pi(h, s, a) + p(h, s) \right)$$

$$+ \sum_{a',s'':P_{|h|+1}(s,a',s'')>0} \sum_{h',s':(h+(s,a'),s'')\subseteq(h',s')} \left( \sum_{a\in\mathcal{A}} v^P_{|h'|+1}(s', a) \cdot x(h', s', a) + y(h', s') \right)$$

By the induction hypothesis, the second sum above is equal to

$$\sum_{a',s'':P_{|h|+1}(s,a',s'')>0} \text{obj}(h + (s, a'), s'')$$

$$= \sum_{a',s'':P_{|h|+1}(s,a',s'')>0} z(h + (s, a'), s'') \cdot u^M_P(h + (s, a'), s'')$$

$$= \sum_{a',s'':P_{|h|+1}(s,a',s'')>0} x(h, s, a') \cdot P^E_{|h|+1}(s, a', s'') \cdot u^M_P(h + (s, a'), s'')$$

$$= \sum_{a\in\mathcal{A},s'\in\mathcal{S}} x(h, s, a) \cdot P_{|h|+1}(s, a, s') \cdot u^M_P(h + (s, a), s')$$

$$= z(h, s) \cdot \sum_{a\in\mathcal{A}} \left( \pi(h, s, a) \cdot \sum_{s'\in\mathcal{S}} P_{|h|+1}(s, a, s') \cdot u^M_P(h + (s, a), s') \right).$$

Putting this back into the above expression for $\text{obj}(h, s)$, we get

$$\text{obj}(h, s)$$

$$= z(h, s) \cdot \left( \sum_{a\in\mathcal{A}} v^P_{|h|+1}(s, a) \cdot \pi(h, s, a) + p(h, s) \right)$$

$$+ z(h, s) \cdot \sum_{a\in\mathcal{A}} \left( \pi(h, s, a) \cdot \sum_{s'\in\mathcal{S}} P_{|h|+1}(s, a, s') \cdot u^M_P(h + (s, a), s') \right)$$

$$= z(h, s) \cdot \left( \sum_{a\in\mathcal{A}} \cdot\pi_{|h|+1}(h, s, a) \cdot \left( v^P_{|h|+1}(s, a) + \sum_{s'\in\mathcal{S}} P_{|h|+1}(s, a, s') \cdot u^M_P(h + (s, a), s') \right) + p(h, s) \right)$$

$$= z(h, s) \cdot u^M_P(h, s).$$

So for any $h \in \mathcal{H}$, $s \in \mathcal{S}$, $\text{obj}(h, s) = z(h, s) \cdot u^M_P(h, s)$. Then we immediately have

$$u^M_P(\emptyset) = \sum_{s\in\mathcal{S}} P_0(s) \cdot u^M_P(\emptyset, s) = \sum_{s\in\mathcal{S}} z(\emptyset, s) \cdot u^M_P(\emptyset, s) = \sum_{s\in\mathcal{S}} \text{obj}(\emptyset, s) = \text{obj}. \qquad \square$$

*Proof of Lemma 3.* By Eq. (4) and Lemma 2, for all $h, s, s'$,

$$u(h, s, s')$$

$$= \sum_{a \in \mathcal{A}} v_{|h|+1}^A(s, a) \cdot x(h, s', a) - y(h, s')$$

$$+ \sum_{a \in \mathcal{A}, s'' \in \mathcal{S}} \frac{P_{|h|+1}(s, a, s'')}{P_{|h|+1}^E(s', a, s'')} \cdot z(h + (s', a), s'') \cdot u_A^M(h + (s', a), s'') \qquad \text{(Lemma 2)}$$

$$= \sum_{a \in \mathcal{A}} v_{|h|+1}^A(s, a) \cdot x(h, s', a) - y(h, s') + \sum_{a \in \mathcal{A}, s'' \in \mathcal{S}} P_{|h|+1}(s, a, s'') \cdot x(h, s', a) \cdot u_A^M(h + (s', a), s'')$$

$$\text{(Eq. (4))}$$

Now by rearranging the above expression and applying the construction of the mechanism $M$ and the single-step reporting strategy $r_{h,s,s'}$, we have

$$u(h, s, s')$$

$$= \sum_{a \in \mathcal{A}} x(h, s', a) \left( v_{|h|+1}^A(s, a) + \sum_{s'' \in \mathcal{S}} P_{|h|+1}(s, a, s'') \cdot u_A^M(h + (s', a), s'') \right) - y(h, s')$$

$$\text{(rearranging)}$$

$$= z(h, s') \cdot \left( \sum_{a} \pi(h, s', a) \cdot \left( v_{|h|+1}^A(s, a) + \sum_{s''} P_{|h|+1}(s, a, s'') \cdot u_A^M(h + (s', a), s'') \right) - p(h, s') \right)$$

$$\text{(construction of mechanism)}$$

$$= z(h, s') \cdot \left( \sum_{a} \pi(h, s', a) \cdot \left( v_{|h|+1}^A(s, a) + \sum_{s''} P_{|h|+1}(s, a, s'') \cdot u_A^{M, r_{h,s,s'}}(h + (s', a), s'') \right) - p(h, s') \right)$$

$$\text{(construction of } r_{h,s,s'})$$

$$= z(h, s') \cdot u_A^{M, r_{h,s,s'}}(h, s), \qquad\qquad \text{(definition of } u_A^{M, r_{h,s,s'}})$$

as desired. $\qquad\qquad\qquad\qquad\qquad\qquad\qquad\qquad\qquad\qquad\qquad\qquad\qquad\qquad\qquad\qquad\square$

*Proof of Lemma 4.* Fix $h \in \mathcal{H}$, $s, s' \in \mathcal{S}$, and let $(s_p, a_p) = \text{last}(h)$. When $h = \emptyset$, by Lemmas 2 and 3 and Eq. (3),

$$u(h, s) \geq \frac{P_{|h|}^E(s_p, a_p, s)}{P_{|h|}^E(s_p, a_p, s')} \cdot u(h, s, s')$$

$$\iff z(\emptyset, s) \cdot u_A^M(\emptyset, s) \geq \frac{P_0^E(s_p, a_p, s)}{P_0^E(s_p, a_p, s')} \cdot z(\emptyset, s') \cdot u_A^{M, r_{\emptyset, s, s'}}(\emptyset, s)$$

$$\iff z(\emptyset, s) \cdot u_A^M(\emptyset, s) \geq \frac{P_0^E(s)}{P_0^E(s')} \cdot z(\emptyset, s') \cdot u_A^{M, r_{\emptyset, s, s'}}(\emptyset, s)$$

$$\iff u_A^M(\emptyset, s) \geq u_A^{M, r_{\emptyset, s, s'}}(\emptyset, s).$$

When $|h| > 0$, suppose $h = (s_1, a_1, \ldots, s_t, a_t)$, and let $h_p = (s_1, a_1, \ldots, s_{t-1}, a_{t-1})$. By Lemmas 2 and 3 and Eq. (2),

$$u(h, s) \geq \frac{P_{|h|}^E(s_p, a_p, s)}{P_{|h|}^E(s_p, a_p, s')} \cdot u(h, s, s')$$

$$\iff z(h, s) \cdot u_A^M(h, s) \geq \frac{P_{|h|}^E(s_p, a_p, s)}{P_{|h|}^E(s_p, a_p, s')} \cdot z(h, s') \cdot u_A^{M, r_{h, s, s'}}(h, s)$$

$$\iff x(h_p, s_p, a_p) \cdot u_A^M(h, s) \geq x(h_p, s_p, a_p) \cdot u_A^{M, r_{h, s, s'}}(h, s).$$

Note that when $x(h_p, s_p, a_p) = 0$, $(h, s)$ cannot be reached, because (1) if $z(h_p, s_p) > 0$, then when the (reported) history-state pair is $(h_p, s_p)$, the mechanism never takes action $a_p$, and (2) if $z(h_p, s_p) = 0$, then such an impossible action exists somewhere in $h_p$. In such cases, $\pi(h, s)$ and

$p(h, s)$ will never be accessed, since it is impossible for the (reported) history to be $h$. In other words, when $(h, s)$ is reachable, we must have $x(h_p, s_p, a_p) > 0$, in which case the last inequality is equivalent to $u_A^M(h, s) \geq u_A^{M, r_{h,s,s'}}(h, s)$. $\qquad \square$

*Proof of Lemma 5.* We only need to show that IC is equivalent to robustness against single-step misreporting. We prove this inductively, aiming to eliminate misreporting one step at a time. To be more specific, consider the following partial reporting strategy. For a reporting strategy $r$, $t \in [T]$, let $r|_{\geq t}$ denote the reporting strategy restricted to time $t, t+1, \ldots, T$, i.e., for any $h' \in \mathcal{H}$, $s' \in \mathcal{S}$,

$$r|_{\geq t}(h', s') = \begin{cases} s', \text{if } |h'| + 1 < t \\ r(h', s'), \text{otherwise} \end{cases}.$$

Similarly, let $r|_{<t}$ denote $r$ restricted to time $1, 2, \ldots, t-1$, and $r|_{=t}$ denote $r$ restricted to time $t$. We show inductively that for any reachable history-state pair $(h, s)$, and any reporting strategy $r$,

$$u_A^{M, (r|_{<|h|+1})}(h, s) \geq u_A^{M, r}(h, s).$$

Without loss of generality, we assume that for any unreachable pair $(h', s')$, $r$ simply reports truthfully, i.e., $r(h', s') = s'$.

Recall that $r(h)$ is the reported history given by $r$ when the true history is $h$. When $|h| = T - 1$, the above claim is implied by Lemma 4, because

$$u_A^{M, r}(h, s) = u_A^{M, (r|_{\geq T})}(r(h), s) \geq u_A^M(r(h), s) = u_A^{M, (r|_{<T})}(h, s).$$

Now suppose $|h| < T - 1$. By the induction hypothesis, we have

$$u_A^{M, r}(h, s) = u_A^{M, (r|_{\geq |h|+1})}(r(h), s) \leq u_A^{M, ((r|_{\geq |h|+1})|_{<|h|+2})}(r(h), s) = u_A^{M, (r|_{=|h|+1})}(r(h), s).$$

Now again by Lemma 4, we have

$$u_A^{M, r}(h, s) \leq u_A^{M, (r|_{=|h|+1})}(r(h), s) \leq u_A^M(r(h), s) = u_A^{M, (r|_{<|h|+1})}(h, s),$$

which establishes the above claim.

Now observe that as a special case of the claim, for any $s \in \mathcal{S}$,

$$u_A^{M, r}(\emptyset, s) \leq u_A^{M, (r|_{<1})}(\emptyset, s) = u_A^M(\emptyset, s).$$

Now summing over $s$, this implies that for any reporting strategy $r$,

$$u_A^{M, r}(\emptyset) = \sum_{s \in \mathcal{S}} P_0(s) \cdot u_A^{M, r}(\emptyset, s) \leq \sum_{s \in \mathcal{S}} P_0(s) \cdot u_A^M(\emptyset, s) = u_A^M(\emptyset). \qquad \square$$

*Proof of Theorem 2.* Given the correspondence between mechanisms and LP variables, by Lemma 5, it is easy to see that (modulo the unreachable parts) every IC and IR mechanism corresponds bijectively to a feasible solution to the LP in Figure 1. Moreover, by Lemma 1, the objective value of this solution is precisely the principal's overall utility, which directly implies that an optimal solution to the LP corresponds to an IC and IR mechanism which maximizes the principal's overall utility.

Now observe that the number of variables and the number of constraints in the LP are both $O(|\mathcal{S}|^{T+1}|\mathcal{A}|^T)$. Moreover, all relevant coefficients in the LP can be encoded using $O(L)$ bits. It is well-known that such an LP can be solved in time $\text{poly}(|\mathcal{S}|^T, |\mathcal{A}|^T, L)$. $\qquad \square$

## H   Omitted Proofs from Section D

*Proof of Lemma 7.* We construct $M'$ explicitly based on $M$. Let $\pi'(t, s_p, a_p, s, a)$ be the probability that $M'$ chooses action $a$ at time $t$ in state $s$ when the previous state-action pair is $(s_p, a_p)$. Similarly, let $p'(t, s_p, a_p, s)$ be the payment specified by $M'$ at time $t$ in state $s$ when the previous state-action

pair is $(s_p, a_p)$. We construct $M'$ from $M$ inductively as follows. For each $t \in [T]$, $s_p \in \mathcal{S}$ and $a_p \in \mathcal{A}$, let $h^*(t, s_p, a_p) \in \mathcal{H}_{t-1}$ be any history such that

$$h^*(t, s_p, a_p) \in \operatorname*{argmax}_{h \in \mathcal{H}_{t-1}:(s_p,a_p)=\mathrm{last}(h)} \sum_{s \in \mathcal{S}} P_{t-1}(s_p, a_p, s) \cdot \left( p(h, s) + \sum_{a \in \mathcal{A}} \pi(h, s, a) \cdot \left( v^P_{|h|+1}(s, a) \right.\right.$$

$$\left.\left. + \sum_{s' \in \mathcal{S}} P_t(s, a, s') \cdot u^M_P(h + (s, a), s') \right) \right).$$

Then, for all $s \in \mathcal{S}$, let

$$\pi'(t, s_p, a_p, s) = \pi(h^*(t, s_p, a_p), s) \quad \text{and} \quad p(t, s_p, a_p, s) = p(h^*(t, s_p, a_p), s).$$

This finishes the construction of $M'$.

We first show that $u^{M'}_P(\emptyset) \geq u^M_P(\emptyset)$, by inductively showing a stronger claim: for all $h \in \mathcal{H}$,

$$\sum_s P_{|h|}(s_p, a_p, s) \cdot u^{M'}_P(h, s) \geq \sum_s P_{|h|}(s_p, a_p, s) \cdot u^M_P(h, s),$$

where $(s_p, a_p) = \mathrm{last}(h)$. For all $h \in \mathcal{H}_{T-1}$, letting $(s_p, a_p) = \mathrm{last}(h)$, by the construction of $M'$, we have

$$\sum_s P_{T-1}(s_p, a_p, s) \cdot u^{M'}_P(h, s) = \sum_s P_{T-1}(s_p, a_p, s) \cdot u^M_P(h^*(T, s_p, a_p), s)$$

$$\geq \sum_s P_{T-1}(s_p, a_p, s) \cdot u^M_P(h, s).$$

Now for all $h \in \mathcal{H}$ where $|h| < T - 1$, letting $(s_p, a_p) = \mathrm{last}(h)$ and $h^* = h^*(|h| + 1, s_p, a_p)$, we have

$$\sum_s P_{|h|}(s_p, a_p, s) \cdot u^{M'}_P(h, s)$$

$$= \sum_s P_{|h|}(s_p, a_p, s) \cdot \left( p(h^*, s) + \sum_{a \in \mathcal{A}} \pi(h^*, s, a) \cdot \left( v^P_{|h|+1}(s, a) + \sum_{s' \in \mathcal{S}} P_t(s, a, s') \cdot u^{M'}_P(h + (s, a), s') \right) \right)$$

$$= \sum_s P_{|h|}(s_p, a_p, s) \cdot \left( p(h^*, s) + \sum_{a \in \mathcal{A}} \pi(h^*, s, a) \cdot \left( v^P_{|h|+1}(s, a) + \sum_{s' \in \mathcal{S}} P_t(s, a, s') \cdot u^{M'}_P(h^* + (s, a), s') \right) \right)$$
$$\text{(property of } M')$$

$$\geq \sum_s P_{|h|}(s_p, a_p, s) \cdot \left( p(h^*, s) + \sum_{a \in \mathcal{A}} \pi(h^*, s, a) \cdot \left( v^P_{|h|+1}(s, a) + \sum_{s' \in \mathcal{S}} P_t(s, a, s') \cdot u^M_P(h^* + (s, a), s') \right) \right)$$
$$\text{(induction hypothesis)}$$

$$\geq \sum_s P_{|h|}(s_p, a_p, s) \cdot \left( p(h, s) + \sum_{a \in \mathcal{A}} \pi(h, s, a) \cdot \left( v^P_{|h|+1}(s, a) + \sum_{s' \in \mathcal{S}} P_t(s, a, s') \cdot u^M_P(h + (s, a), s') \right) \right)$$
$$\text{(choice of } h^*)$$

$$= \sum_s P_{|h|}(s_p, a_p, s) \cdot u^M_P(h, s).$$

Then in particular, we have

$$u^{M'}_P(\emptyset) = \sum_s P_0(s) \cdot u^{M'}_P(\emptyset, s) \geq \sum_s P_0(s) \cdot u^M_P(\emptyset, s) = u^M_P(\emptyset).$$

Finally we prove that $M'$ is IC. By the proof of Lemma 5, we only need to show that $M'$ is robust against any single-step reporting strategy $r_{h,s,s'}$. In fact, letting $(s_p, a_p) = \mathrm{last}(h)$ and $h^* = h^*(|h| + 1, s_p, a_p)$,

$$u^{M'}_A(h, s) = \sum_a \pi(h^*, s, a) \cdot v^A_{|h|+1}(s, a) + p(h^*, s) = u^M_A(h^*, s).$$

Moreover,

$$u_A^{M',r_{h,s,s'}}(h,s) = \sum_a \pi(h^*,s',a) \cdot v_{|h|+1}^A(s,a) + p(h^*,s) = u_A^{M,r_{h,s,s'}}(h^*,s).$$

Since $M$ is IC, we have

$$u_A^{M'}(h,s) = u_A^M(h^*,s) \geq u_A^{M,r_{h,s,s'}}(h^*,s) = u_A^{M',r_{h,s,s'}}(h,s).$$

Now by the argument in the proof of Lemma 5, we know that for all reporting strategy $r$, $h \in \mathcal{H}$, $s \in \mathcal{S}$,

$$u_A^{M',r}(h,s) \leq u_A^{M',(r|_{<|h|+1})}(h,s),$$

so

$$u_A^{M',(r|_{\geq|h|+1})}(h,s) \leq u_A^{M',((r|_{\geq|h|+1})|_{<|h|+1})}(h,s) = u_A^{M'}(h,s),$$

which is precisely the IC requirement for myopic agents. Similar arguments guarantee that $M'$ has the same IR property as $M$. $\qquad\square$

*Proof of Theorem 4.* We first argue the easy part, i.e., the time complexity. Observe that calls to OptStatMech dominates the time complexity. Moreover, the algorithm makes $T|\mathcal{S}||\mathcal{A}|$ calls to OptStatMech, so the overall time complexity is as stated.

Now we show the optimality of the computed mechanism $M$. We prove inductively a stronger claim, i.e., for any $t \in [T]$, $s_p \in \mathcal{S}$, $a_p \in \mathcal{A}$,

$$\sum_s P_0(s_p,a_p,s) \cdot u_P^M(t,s_p,a_p,s) = \max_{M'} P_0(s_p,a_p,s) \cdot u_P^{M'}(t,s_p,a_p,s),$$

where the maximum is over all succinct mechanisms $M'$ that are IC and (optionally) IR. First observe that for all $s \in \mathcal{S}$, $a \in \mathcal{A}$,

$$u(T,s,a) = v_T^P(s,a).$$

So, for all $s_p \in \mathcal{S}$, $a_p \in \mathcal{A}$,

$$\sum_s P_0(s_p,a_p,s) \cdot u_P^M(T,s_p,a_p,s)$$

$$= \sum_s P_0(s_p,a_p,s) \cdot \left( p(T,s_p,a_p,s) + \sum_a \pi(T,s_p,a_p,s,a) \cdot v_T^P(s,a) \right)$$

$$= \max_{M'=(\pi',p')} \sum_s P_0(s_p,a_p,s) \cdot \left( p'(T,s_p,a_p,s) + \sum_a \pi'(T,s_p,a_p,s,a) \cdot v_T^P(s,a) \right)$$

(optimality of $M$ at time $T$ as a static mechanism)

$$= \max_{M'} \sum_s P_0(s_p,a_p,s) \cdot u_P^M(T,s_p,a_p,s).$$

Again, the maximum is over all succinct mechanisms $M'$ that are IC and (optionally) IR.

Now for $t \in [T-1]$, by the construction of $M$,

$$\sum_s P_0(s_p,a_p,s) \cdot u_P^M(t,s_p,a_p,s)$$

$$= \sum_s P_0(s_p,a_p,s) \cdot \left( p(t,s_p,a_p,s) + \sum_a \pi(t,s_p,a_p,s,a) \cdot \left( v_t^P(s,a) \right.\right.$$

$$\left.\left. + \sum_{s'} P_t(s,a,s') \cdot u_P^M(t+1,s,a,s') \right) \right)$$

$$= \max_{M'=(\pi',p')} \sum_s P_0(s_p,a_p,s) \cdot \left( p'(t,s_p,a_p,s) + \sum_a \pi'(t,s_p,a_p,s,a) \cdot \left( v_t^P(s,a) \right.\right.$$

$$\left.\left. + \sum_{s'} P_t(s,a,s') \cdot u_P^M(t+1,s,a,s') \right) \right). \quad \text{(optimality of } M \text{ at time } t \text{ as a static mechanism)}$$

By the induction hypothesis and the fact that $M'$ is succinct,

$$\sum_s P_0(s_p, a_p, s) \cdot u_P^M(t, s_p, a_p, s)$$

$$= \max_{M'=(\pi',p')} \sum_s P_0(s_p, a_p, s) \cdot \left( p'(t, s_p, a_p, s) + \sum_a \pi'(t, s_p, a_p, s, a) \cdot \left( v_t^P(s, a) \right. \right.$$

$$\left. \left. + \max_{M''} \sum_{s'} P_t(s, a, s') \cdot u_P^{M''}(t+1, s, a, s') \right) \right) \qquad \text{(induction hypothesis)}$$

$$= \max_{M'=(\pi',p')} \sum_s P_0(s_p, a_p, s) \cdot \left( p'(t, s_p, a_p, s) + \sum_a \pi'(t, s_p, a_p, s, a) \cdot \left( v_t^P(s, a) \right. \right.$$

$$\left. \left. + \sum_{s'} P_t(s, a, s') \cdot u_P^{M'}(t+1, s, a, s') \right) \right) \qquad (M' \text{ is succinct})$$

$$= \max_{M'} \sum_s P_0(s_p, a_p, s) \cdot u_P^{M'}(t, s_p, a_p, s).$$

All maxima are over all succinct mechanisms that are IC and (optionally) IR. As a result, we have

$$u_P^M(\emptyset) = \sum_s P_0(s) \cdot u_P^M(\emptyset, s) = \max_{M'} \sum_s P_0(s) \cdot u_P^{M'}(\emptyset, s) = \max_{M'} u_P^{M'}(\emptyset). \qquad \square$$

# I  Omitted Proofs from Section E

*Proof of Theorem 5.* First suppose the agent is patient and without loss of generality has a discount factor of 1. Let $T = 2$ and $\mathcal{S} = \mathcal{A} = [n]$ where $n \geq \varepsilon^{-1}$. The initial distribution is uniform over $[n]$, i.e., $P_0(i) = 1/n$ for all $i \in [n]$, i.e., no matter what action is played, all states always transition to state 1. The transition operator is such that $P_1(i, j, 1) = 1$ for all $i, j \in [n]$. At time $T = 2$, the principal's valuations are $v_T^P(i, j) = 0$ for all $i, j \in [n]$. At time 1, the principal's valuation function is such that for all $i, j \in [n]$, $v_1^P(i, j) = 1$ if $i = j$, and $v_1^P(i, j) = 0$ if $i \neq j$. For $t \in [T]$, the agent's valuation function is such that for all $i, j \in [n]$, $v_t^A(i, j) = 0$ if $i = j$, and $v_t^A(i, j) = 1$ if $i \neq j$.

Consider the principal's optimal utility, which is clearly upper bounded by 1 (1 at time 1 and 0 at time 2). The following mechanism is IC and achieves this upper bound:

- At time 1, play action $i$ for each state $i \in [n]$.

- At time $T = 2$, play action $(i \bmod n) + 1$ iff the state at time 1 is $i$.

The mechanism is IC because regardless of the (reported) initial state, the agent achieves overall utility 1. It is easy to check this mechanism achieves utility 1.

On the other hand, any memoryless IC mechanism can achieve utility at most $1/n \leq \varepsilon$. This is because at time $T = 2$, the current state provides absolutely no information, so the mechanism has to perform the same (randomized) action regardless of the initial state. As a result, in order to be IC, the mechanism has to satisfy the following condition at time 1: for all $i, j \in [n]$, $\pi(i, i) \leq \pi(j, i)$, where $\pi(a, b)$ is the probability that action $b$ is played in state $a$ at time 1. So the principal's utility can be bounded as follows:

$$\frac{1}{n} \sum_i \pi(i, i) \leq \frac{1}{n} \sum_i \left( \frac{1}{n} \sum_j \pi(j, i) \right) = \frac{1}{n^2} \sum_{i,j} \pi(j, i) = \frac{1}{n}.$$

This concludes the proof when the agent is patient.

Now consider the case with a myopic agent. Again, let $T = 2$ and $\mathcal{S} = \mathcal{A} = [n]$ where $n \geq \varepsilon^{-1}$. The initial distribution is again uniform over $[n]$, i.e., $P_0(i) = 1/n$ for all $i \in [n]$. The transition operator is such that $P_1(i, j, i) = 1$ for all $i, j \in [n]$, i.e., no matter what action is played, state $i$ always transitions to state $i$. At time 1, the principal's and the agent's valuations are $v_1^P(i, j) = v_1^A(i, j) = 0$

for all $i, j \in [n]$. At time $T = 2$, the principal's valuation function is such that for all $i, j \in [n]$, $v_T^P(i, j) = 1$ if $i = j$, and $v_T^P(i, j) = 0$ if $i \neq j$. And the agent's valuation function is such that for all $i, j \in [n]$, $v_T^A(i, j) = 0$ if $i = j$, and $v_T^A(i, j) = 1$ if $i \neq j$.

The principal's optimal utility, 1, is achieved by the following succinct (but not memoryless) IC mechanism:

- At time 1, play action 1 for all states.

- At time 2, play action $i$ iff the state at time 1 is $i$.

The mechanism is IC in particular because the agent is myopic and cannot change the past. It is easy to check this mechanism achieves utility 1.

On the other hand, any memoryless IC mechanism can achieve utility at most $1/n \leq \varepsilon$. This is because at time $T = 2$, the mechanism cannot memorize anything before, so it has to be IC based only on the current state, which puts the mechanism in a situation that is essentially the same as at time 1 in the hard instance for patient agents. Similar arguments then guarantee that the principal's utility is at most $1/n$, which concludes the proof for myopic agents. Finally, we note that the above constructions work even if payments are allowed. $\qquad\square$

*Proof of Theorem 6.* We use reductions from MAX-SAT similar to that in Theorem 1 for both myopic and patient agents. First consider the case where the agent is patient with a discount factor of 1. In this case, the reduction in Theorem 1 applies without any modification. In particular, since the principal and the agent are in a zero-sum situation, without loss of generality, any optimal memoryless mechanism does not depend on the reported states. And again, since the principal's utility is multilinear in the actions at each time, there is a deterministic mechanism which is optimal. As argued in the proof of Theorem 1, such a mechanism corresponds precisely to an optimal assignment of variables in the MAX-SAT instance, which implies the $7/8 + \varepsilon$ inapproximability.

Now consider the case where the agent is myopic. Here we slightly modify the reduction, and in particular, the agent's valuation functions. That is, for each $t \in [T]$ and $i \in [m]$, we let

$$v_t^A(s, a_{\mathrm{pos}}) = c \quad \text{and} \quad v_t^A(s, a_{\mathrm{neg}}) = 0,$$

for all $s \in \mathcal{S}$, where $c > 0$ is an arbitrarily small constant. This guarantees that at any time $t$, in order to be IC, the (randomized) actions for all states have to be exactly the same. Then since the principal's utility is multilinear, again it is without loss of generality to consider deterministic mechanisms, which correspond to assignments of variables. The ratio of $7/8 + \varepsilon$ follows immediately. Finally, we remark that the above reductions still work when payments are allowed. $\qquad\square$