# OpenReview forum: "Automated Dynamic Mechanism Design"
_NeurIPS.cc/2021/Conference — NeurIPS 2021 Poster_

### Official Review · Reviewer_bSJY · 2021-07-03

**Rating:** 8
**Confidence:** 3

**Summary:**

The authors formulate and study dynamic mechanism design in an MDP-like setting. The motivation is principal-agent problems. The principal commits to a policy to choose actions given observations of states; the environment evolves according to some transitions dynamics based on these actions; the agent does not act but is free to misreport states to the principal. Agent and principal receive utility; the principal’s goal is to choose a policy that is incentive compatible and individually rational (standard goals of mechanism design) while maximizing its own expected total utility.

In general, from a full description of the environment and dynamics, the “planning” type problem to find an optimal mechanism is NP-hard (shown via reduction from MAXSAT). If the time horizon of the problem is treated as a constant, then it is possible to formulate a linear program to find the optimal mechanism in polynomial time, and doing so is a major portion of the paper.


**Limitations And Societal Impact:**

This is a highly theoretical paper and any negative social impacts would be very remote. As for limitations, the authors do discuss them when relevant (e.g. the results require assuming that principal utility must depend linearly on payments, if the revelation principle doesn't hold in a static setting then it won't hold for dynamic). Moreover a lot of the paper is about computational hardness results which represent a kind of limitation.

**Main Review:**

## Longer summary

The motivation is a principal-agent problem. The principal is going to commit to some policy mapping world states to (distributions over) actions; then the agent can choose to lie about the state of the world; based on this possibly false state, the principal’s policy acts, and the agent and the principal both receive some utility based on states and actions. In the single-step case, this is a fairly general description of mechanism design with a single agent. Then the standard goals of the principal are to maximize its own utility, subject to the incentive compatibility (IC) constraint that agents should not gain utility from lying, and the individual rationality (IR) constraint that agents should be assured they will never have negative utility.

To generalize to the multi-step case, the authors define an MDP-like environment which transitions from state to state over multiple time horizons. The dynamics of this environment are Markovian, in that transitions only depend on the current state and action; however both principal and agent are aware of their full histories. Both principal and agent receive immediate utility based on the current state and action, and their goal is to maximize total expected utility over future time steps. The definition of IC is generalized to the dynamic setting to allow for misreporting strategies that depend on the whole history of states. Likewise the notion of IR is generalized in two ways: either guaranteeing that expected utility at the start is positive, or guaranteeing that expected forward-looking utility never goes negative (so the agent will never leave the mechanism).

This formalization of dynamic mechanism design is carefully and clearly done. A major part of the paper is then a linear program which, given a description of the environment, maximizes principal utility subject to constraints enforcing IR and IC for the agent. It’s natural to notice that this program has constraints for each possible state-action history, which grows exponentially with the length of the episode, so it’s only really poly-time if the time horizon is treated as a constant. The authors address this point and show via a reduction from MAXSAT that this is unavoidable — for arbitrarily long time horizons the problem is inevitably NP hard even to approximate.

Other points worth mentioning:

- Markovian policies, which in regular MDPs are optimal, may be arbitrarily far from optimal. Furthermore, in this setting restricting to Markovian policies doesn’t even make things easier to compute (the MAXSAT reduction still works). This is an interesting result.
- The general problem is NP hard. By treating time horizon as a constant one can avoid this (and formulate the LP). It’s also possible to avoid this if the agent is restricted to be myopic. In this case, optimal policies only require the current state and 1 step of history, which allows the authors to implement a dynamic-programming-style algorithm which is faster than solving the general LP.

There are also some experiments in which the authors actually implement their LPs for a simple setting.

## Evaluation

### Originality

The paper carefully extends existing settings for mechanism design to a dynamic setting, and considers many special cases and design goals. The results all tend to come out as expected, and the techniques seem fairly familiar, but I wouldn’t call it unoriginal.

### Quality

The paper is high-quality with a lot of attention to detail.

### Clarity

The paper is clearly written and easy to understand. Due to the unfortunate space limitations, a lot has been moved to the appendix, which fragments the paper a little. I liked the introduction but I think it could have been condensed a lot in order to squeeze more appendix material into the main body, and this would have helped the paper.

### Significance

The paper provides a useful framework for thinking about dynamic mechanism design, and the results about computational hardness are relevant, especially in comparison to single-agent MDPs. I think the formalization here could end up being a very useful starting point for people who want to study new techniques for this type of mechanism design problem, including e.g. extending reinforcement learning techniques to work in this setting.

## Questions for authors

### Misreporting strategies

Can the agent be inconsistent in their misreports? In other words, can they misreport s_t as s_t’ one round, and then s_t’’ the next round? It seems like the answer is no due to the definition of the misreporting strategy. But perhaps spelling this out explicitly might be useful.

### Myopic agents and memoryless mechanisms

Theorem 3 says that computing an optimal general mechanism for myopic agents can be done in polynomial time (because a “succinct” two-step-history mechanism will be optimal). Theorem 5 says that computing an optimal memoryless mechanism, including for myopic agents, is hard to approximate. Is there some intuition of how just going from general (actually succinct) mechanisms, to memoryless mechanisms, makes the problem computationally harder? It’s possible I’ve misunderstood some assumptions of these two theorems.


Final update: the authors helpfully answered my questions and my opinion of the paper is still quite positive.




**Time Spent Reviewing:**

5

---

> ### Author Response · Authors · 2021-08-10
> **Author Response to Reviewer bSJY**
>
> Thank you for your thoughtful and helpful comments.
>
> Re presentation: we will at least make a full version of the paper available, where hopefully the results can be arranged in a more enjoyable way.
>
> Re misreporting strategies: the reviewer is right that the current setup does not allow this type of inconsistency.  Conceptually, this is because the principal can simply remember previous reports, so there is no point in misreporting about the history.  We will make this clear.  (On the other hand, the current model does allow for "impossible" reports, i.e., those happen with probability 0.)
>
> Re myopic agents and memoryless mechanisms: the "succinct" mechanisms (as the reviewer points out) do have memory for the previous state and action.  Intuitively, this provides information about the roll-in distribution at the current time, which is crucial for optimizing for the optimal mechanism.  Memoryless ones, on the other hand, can only depend on the current state.  So in general, optimal memoryless mechanisms can provide strictly worse utility for the principal, and the structure of optimal memoryless mechanisms can be quite different from those with just a bit more memory.  The hardness result simply says it's hard to optimize within this restricted class of mechanisms.

---

### Official Review · Reviewer_KjrC · 2021-07-16

**Rating:** 7
**Confidence:** 3

**Summary:**

The authors study the problem of computing optimal mechanisms in a dynamic unstructured environment. Unlike a static environment, the principal is allowed to repeatedly interact with a strategic agent and take actions based on the agent's reports of the current state of the world. The authors show that when the time horizon of such a dynamic environment is not finite, the problem is intractable. When the time horizon is small, they show that an efficient mechanism can be computed using an LP

**Main Review:**

- The contribution is significantly novel - much work in the AMD literature has been on the static environment. Additionally, since the environment that the authors consider is also only loosely structured, their method differs from the typical characterize-and-solve approach to mechanism design.

- The claims seem to be technically sound however I didn't check the proofs in detail.

- The paper is well written/ clear

- The authors provide a compelling example in the introduction as to why studying dynamic mechanism design for unstructured environments is significant. I believe it's a problem of interest to people in both the ML and the mechanism design community.



**Time Spent Reviewing:**

3 hours

---

> ### Author Response · Authors · 2021-08-10
> **Author Response to Reviewer KjrC**
>
> Thank you for your thoughtful and helpful comments.

---

### Official Review · Reviewer_AaYK · 2021-07-16

**Rating:** 7
**Confidence:** 2

**Summary:**

This paper presents and proves a linear program for unstructured dynamic mechanism design. The LP supports payments and different individual-rationality constraints. The key contributions are:
1) Presenting an LP that provably yields an optimal mechanism for an unstructured dynamic environment with a finite time horizon.
2) From this LP formulation and standard LP complexity results follows a polynomial runtime guarantee for finding an optimal mechanism.

**Limitations And Societal Impact:**

The authors did not discuss potential negative societal impact, but the research is very fundamental so I do not think it is necessary. The limitations are also discussed (e.g. constant horizon).

**Main Review:**

1. This work addresses a novel task of computing optimal mechanisms in unstructured dynamic environments. Previous works have explored automated mechanism design for static environments, where agents make a memoryless single decision rather than repeated interactions with the same mechanism. This makes the paper the first such result for dynamic environments, which is of interest to the broader Economics and Computation community.
2. The paper is well written and organized. The build up of the linear program (Figure 1) through lemmas 1-5 was very clear. However, it may be helpful to leave additional space for discussion of the main result (Theorem 2). As someone unfamiliar with the analogous results for non-dynamic settings, it is not immediately clear how to interpret the upper bound: were we expecting the complexity to potentially be non-polynomial? is there potential to improve on this bound (e.g. weaken the exponential dependence on T)?
3. As the paper is purely theoretical, it is not clear how practical solving the LP is in experimental settings. The theoretical results are significant in that they’re the first such result for dynamic environments. The hardness result (Theorem 1) is not surprising but still an important justification for the paper’s focus on constant horizons. The main result (Theorem 2) is important as it shows the LP can be solved in polynomial time, although the complexity bound does feel a bit loose given the exponential dependence on T which could pose a practical issue. The proof technique of setting up an LP is not novel/significant/unique in itself, but it is clean and does the job of showing a loose polynomial complexity bound.
4. Although I only skimmed the proof for Theorem 1 in the Appendix, the MAX-SAT reduction looks sound and the result seems reasonable. The LP construction is correct and Theorem 2 follows straightforwardly from the usual LP guarantees.

*My review is based on the main PDF, and not the full version that is mentioned in the abstract and placed in the supplementary files.

**Time Spent Reviewing:**

6

---

> ### Author Response · Authors · 2021-08-10
> **Author Response to Reviewer AaYK**
>
> Thank you for your thoughtful and helpful comments.
>
> Re interpretation of main algorithmic result: we weren't sure what to expect in dynamic environments.  Some seemingly similar problems (e.g., certain multi-stage strategic classification problems) already become hard if T = 2, but it turns out that's not the case for automated dynamic mechanism design.  As for the dependence on T, the APX-hardness result indicates it's unlikely that the dependence on T can be polynomial, but in principle one may be able to make it subexponential.  Nevertheless, such improvements are impossible if we want to compute a "flat" representation of the mechanism, since the size of such a representation is already exponential in T.
>
> Re practicality of LP-based algorithm: our experiments (Appendix F) show that the algorithm still runs in reasonable time (about 1 hour) for T = 10, and the value of the solution is usually quite close to optimality after 5 or 10 minutes.  With a more careful implementation one may be able to handle up to T = 20, but we wouldn't expect the algorithm to scale well beyond that.

---

> > ### Comment · Reviewer_AaYK · 2021-08-30
> > **Thank you for the clarifications**
> >
> > Thank you for the clarifications about Theorem 2 and the appendix experiments (which I missed in my reading). I've revised my score upwards.

---

### Official Review · Reviewer_D9VS · 2021-07-16

**Rating:** 7
**Confidence:** 3

**Summary:**

This paper studies automated mechanism design in dynamic unstructured dynamic environments. In the setting of interest, a principal and agent repeatedly interact for a finite number of time steps (finite time horizon); at each time step, the agent reports the current state of the environment, the principal takes action, and the environment transitions to the new state. Importantly, the agent's and the principal's utilities are not necessarily the same, implying that the agent might misreport which further implies the need for a mechanism.  The paper considers a randomized dynamic mechanisms that consists of the policy of the principal and the mechanism's payment function. The main results are: a) an LP for finding optimal mechanisms (constant time horizon), b) a computational complexity result which shows that the principal's utility is hard to approximate within a certain factor for environments with large time horizons.  The paper has additional results, as mentioned in the abstract, but these are not presented in the main text.

**Limitations And Societal Impact:**

The checklist indicates that the limitations are discussed, but it is not clear where. The discussion on the limitations could include a discussion on the modeling assumptions (e.g., as those discussed in the main review). Potential negative societal impacts are not discussed, however, in my opinion, this is not needed given the topic of the paper.

**Main Review:**

I enjoyed reading the paper and, to my knowledge, its contributions to the work on automated mechanisms design are novel and significant. My main concerns are related to its structure, quite a few interesting results are in the appendix without actually being presented in the main text. Below I highlight some of the pros and cons of the paper.

Pros:
- This paper is well written and easy to follow, and it clearly explains the setting of interest. The introduction motivates the setting of interest well. That said, the motivating example is more relevant for EconCS than ML.
- The results seem novel, they extend prior work on automated mechanism design to a setting that, to my knowledge, hasn't been studied by prior work.
- The paper provides a relatively complete computational complexity characterization of automated mechanism design for the setting of interest (dynamic unstructured environments, single-agent).
- The most important references (those related to (automated) mechanism design) are adequately presented. Given that the paper studies a principal-agent problem in dynamic environments, a discussion related to stochastic Stackelberg games or environment design could be added.

Cons:
- The paper primarily focuses on the computational complexity aspects of the problem at hand. The learning aspects, which could be more interesting for the NeurIPS community, are not discussed.
- The main algorithmic result, the LP for computing optimal mechanisms, might not be very practical. It assumes that the the transitions probabilities are known and does not scale well with time horizon $T$. While the latter is not surprising (given the hardness result), it might be useful to have some discussion on the former.
- The structure of the paper could be considerably improved. For example, the abstract extensively talks about the results that are not in the main text, including simulation-based experiments, which appear to be quite interesting.

**Time Spent Reviewing:**

4 or 5 hours

---

> ### Author Response · Authors · 2021-08-10
> **Author Response to Reviewer D9VS**
>
> Thank you for your thoughtful and helpful comments.
>
> Re ML aspects of the results: the problem of automated dynamic mechanism design can also be viewed as planning in MDPs with IC and IR constraints.  From this perspective, one may also study, for example, reinforcement learning in such environments (which of course goes beyond the scope of the current paper) -- note that indeed Reviewer bSJY considers our work a very useful starting point for such RL work.
>
> Re related work on stochastic Stackelberg games: we have a paragraph for related work on equilibrium computation (including in various kinds of Stackelberg games) in Appendix B (which we had to move out of the main body due to space limit).  We will try to discuss this briefly in the main paper if there is space, and in any case we will expand the discussion in the full version of the paper.
>
> Re missing discussion on the learning aspect: we will discuss related problems with a stronger learning flavor (e.g., reinforcement learning with IC and / or IR constraints) as future directions.
>
> Re practicality of the LP-based algorithm: we will discuss cases where the transition probabilities are unknown (e.g., in RL tasks).  As for the exponential dependence on T, our experiments (Appendix F) show that the algorithm still runs in reasonable time (about 1 hour) for T = 10, and the value of the solution is usually quite close to optimality after 5 or 10 minutes.  With a more careful implementation one may be able to handle up to T = 20, but we wouldn't expect the algorithm to scale well beyond that.
>
> Re structure of the paper: it is unfortunate that we had to leave a significant number of results out of the main body given the space limit.  We will at least make a full version of the paper available, where hopefully all interesting results can be arranged in a more enjoyable way.

---

### Decision · Program_Chairs · 2021-09-27

**Decision:**

Accept (Poster)

**Comment:**

SUMMARY

The authors consider a dynamic mechanism design problem. In this problem a single agent interacts with a single principal over a time horizon of T rounds. The problem is as follows: There is a distribution over initial states. A start state is drawn from this distribution. In each round, the agent observes the state s and reports a state s'. The principal takes an action a based on the history of states and actions and the reported current state s'. For each state and action pair there is a distribution over next round states, from which the next state is drawn. Both principal and agent have possibly distinct valuation functions that operate on state-action pairs, and quasi-linear utility for money that can flow in either way.

The goal is to design a (possibly randomized) dynamic mechanism that maximizes the principal's utility subject to dynamic IC plus overall or dynamic IR.

The two main results are:

(1) When T is non-constant, it is NP hard to approximate the principal's maximum utility to within a factor of 7/8+eps (Theorem 1).

(2) There is a LP that finds the optimal principal's utility in time O(|S|^T,|A|^T,L) where S is the state space, A is the action space, and L is the number of bits needed to encode each input parameter (LP in Figure 1, Theorem 2).

The paper mentions a couple of other theoretical results including experiments, but all of these are deferred to the appendix.

The theoretical results include an algorithm that scales linearly in T for a myopic agent. A result that shows that memoryless mechanisms, which are optimal for MDPs, do not provide a good solution to the problem in the presence of strategic decisions. The experiments amongst others show that taking incentives into account matters and that optimal designs are remarkably robust against misaligned preferences.

Related work:

The main point of comparison for this paper is a paper by Papadimitriou et al. (SODA 2016 [24]), which considers a closely related mechanism design problem in which the principal's decision in each round is to allocate an item. They show that it is strongly NP hard to complete the optimal deterministic mechanism for a single bidder and two days; and give a LP based algorithm for computing the optimal randomized mechanism when the number of agents and types are both constant (See third comment below).

RECOMMENDATION:

All reviewers liked the motivating story and the and the theoretical results. Although somewhat related, the extension over Papadimitriou et al. seems significant enough to warrant publication in NeurIPS.

For the camera ready:

** I would strongly encourage to cut most of the space used for the LP-based approach (just mention the LP and the theorem with some discussion what the key ingredients are); then use the additional space to state the additional theoretical results and describe some of the experiments

** Also: I think it would be nice to compare in more detail where and why your results are different from Papadimitriou et al., both qualitatively and technically

** Discussion of [24]: You write that [24] give a poly-time LP-based algorithm "when the number of agents and the time horizon are both constant" - I couldn't parse their theorem (Theorem 8 in Section 5 of the arXiv version of that paper). It says "For any number of days D, and a constant number of independent bidders k, the optimal adaptive randomized auction can be found in time polynomial in the number of types and in the number of days."
(please double check that you cite it correctly)

** Related work: The deep learning approach to AMD should be credited to [13].

** Related work: For work on AMD through ML, please also cite:
Payment Rules through Discriminant Based Classifiers
P. Dutting, F. Fischer, P. Jirapinyo, J. K. Lai, B Lubin, D. C. Parkes
ACM EC'12